# Mapping of endosomal proximity proteomes reveals Retromer as a hub for RAB GTPase regulation

Carlos Antón-Plágaro [1] ✉, Kai-en Chen[2], Qian Guo [2], Meihan Liu [2], Ashley J. Evans[1], Philip A. Lewis [3], Kate J. Heesom[3], Kevin A. Wilkinson [4], Brett M. Collins [2] ✉ & Peter J. Cullen [1] ✉

Endosomal retrieval and recycling of integral cargo proteins is essential for cell and organism development and homeostasis and is orchestrated through a specialised endosomal nanodomain, the retrieval sub-domain. Sub-domain dysfunction is associated with human disease, but our mechanistic understanding of its function remains poorly described. Here, using proximity proteomics of retrieval sub-domain components Retromer and Retriever we identify mechanistic details of retrieval sub-domain composition and organization, including an unrecognised complexity in the interface with RAB GTPase switching. Combining X-ray crystallography and in silico predictions with biochemical and cellular analysis, we reveal that Retromer directly associates and recruits the RAB10 regulators DENND4A, DENND4C, TBC1D1, and TBC1D4, and the RAB35 regulator TBC1D13 to regulate retrieval sub-domain function. The retrieval sub-domain therefore constitutes a hub for integrating cargo recycling with the regulated switching of selected RAB GTPases. We propose this constitutes a major component of the neuroprotective role of the retrieval sub-domain.

Across a variety of cell types, three to four thousand transmembrane proteins enter and are sorted and transported through the human endosomal network[1]. Efficient sorting of proteins, including receptors, channels, transporters, polarity cues, and adhesion molecules, is essential for cellular, tissue, and organism-level development and adult homoeostasis and adaptation[1]. An increasing body of evidence points to perturbed endosomal transport being a contributing factor and, in some cases, a causative factor in human diseases, including neurodegenerative and neurological[2–4], and metabolic syndromes and cardiovascular disease[5]. Hijacking the host endosomal network is also an emerging theme in the infectivity of human viral and bacterial pathogens[6].

On entering the endosomal network, transmembrane proteins and their associated proteins and lipids (collectively termed 'cargos') are sorted between two fates. A series of multi-protein ESCRT complexes (endosomal sorting complexes required for transport) recognise cargo modified through monoubiquitylation and transport these for degradation within the lysosome[7]. Other cargos avoid this degradative fate and undergo retrieval for recycling and further rounds of reuse at organelles that include the cell surface, the biosynthetic and autophagic pathways, and lysosomes and lysosome-related organelles[1].

Orchestrating endosomal cargo retrieval and recycling are several multi-protein assemblies that include sorting nexin-27 (SNX27)-Ret-

[1]School of Biochemistry, Faculty of Life Sciences, Biomedical Sciences Building, University of Bristol, Bristol, UK. [2]The University of Queensland, Institute for Molecular Bioscience, St Lucia, Queensland, Australia. [3]Bristol Proteomics Facility, School of Biochemistry, Faculty of Life Sciences, Biomedical Sciences Building, University of Bristol, Bristol, UK. [4]School of Physiology, Pharmacology and Neuroscience, Faculty of Life Sciences, Biomedical Sciences Building, University of Bristol, Bristol, UK. ✉e-mail: carlos.antonplagaro@bristol.ac.uk; b.collins@imb.uq.edu.au; pete.cullen@bristol.ac.uk

romer and SNX3-Retromer[8–16], SNX17-Retriever and associated Commander super-assembly[17–22], the branched F-actin polymerising WASH complex and its regulatory MAGE-L2/USP7/TRIM27 (MUST) complex[23–25], and the SNX-BAR proteins and endosomal sorting complex required for exit-1 (ESCPE-1)[26–31].

On the limiting membrane of endosomes, electron and light microscopy has established that core retrieval and recycling complexes localise to specific retrieval sub-domains that, while present on the same endosome, are physically distinct from ESCRT-demarcated degradative sub-domains[19,32–37]. Analysis in cells and organisms have established the essential importance of functional retrieval sub-domains for development and homeostasis, and disease-associated sub-domain dysregulation is associated with human disease[2–5]. Defining the organisation and function of retrieval sub-domains is central to define the mechanistic basis of endosomal cargo retrieval and recycling.

Although their importance is well established, the precise functional composition of the retrieval sub-domain remains poorly characterised. Here we have employed BioID-based quantitative proteomics to identify the proximity protein environment of the endosomal retrieval sub-domain, specifically focusing on Retromer and Retriever and their respective chief cargo adaptors SNX27 and SNX17 and the HRS component of the ESCRT-0 complex. Through structure led strategic placement of the BioID enzyme, we reveal proximity information relating to the known configuration of Retriever within the Commander super-assembly[20–22] and the arch-like architecture of membrane associated Retromer[14,16]. By establishing molecular components associated with the Retromer demarcated retrieval sub-domain, we identify a previously unrecognised complexity in Retromer's interface with RAB GTPases that we structurally and functionally dissect to reveal the retrieval sub-domain as a major hub for coordinating sequence-dependent cargo recognition and recycling with the regulated switching of a select group of RAB GTPases.

## Results

### Proximity proteomics of endosomal sorting sub-domains

To biochemically probe the organisation of the degradative and retrieval sub-domains we employed the proximity-dependent biotin identification methodology (BioID)[38,39] (Fig. 1A). Upon addition of membrane permeable biotin, the BioID1 enzyme tagged to a protein of interest (POI) catalyses the formation of a highly reactive but labile compound, biotinoyl-5'-AMP, which covalently labels lysine residues of neighbouring proteins within a radius of proximity of approximately 10 nm (for context Retromer is approximately 15–20 nm in length[14,16]). Processing of resulting biotinylated proteins through cell lysis and streptavidin affinity isolation coupled with isobaric tandem mass tagging (TMT) and nanoscale liquid chromatography joined to tandem mass spectrometry (nano-LC-MS/MS) provides an unbiased view of the local proximity proteome (Fig. 1A). Several derivations of BioID are available[40], but we selected BioID1 because of its long record of use and its low basal activity[41], ideal for our CRISPR/Cas9 knock out (KO) and chimeric rescue approach outlined below.

To biochemically probe the organisation of the retrieval sub-domain we focused on: (i) the core VPS35 subunit of Retromer for which we generated two constructs BioID1 tagged at either its carboxy or amino-termini (VPS35-BioID1 and BioID1-VPS35 respectively); (ii) the core VPS35L subunit of Retriever again either carboxy or amino-termini tagged (VPS35L-BioID1 and BioID1-VPS35L respectively); (iii) the major cargo adaptors for Retromer and Retriever, SNX27 and SNX17 respectively, tagged with BioID1 at their amino-termini (BioID1-SNX27 and BioID1-SNX17); and (iv) an amino-terminal tagged version of the ESCRT-0 subunit HRS to label the degradative sub-domain (BioID1-HRS). As additional controls for filtering the selectivity of the BioID1 proximity proteomes we generated HeLa lines expressing a soluble version of cytosolic BioID1 and a version of BioID1 directed to

the cytoplasmic facing mitochondrial external membrane using the TOM70 signal sequence (TOM70-RFP-BioID1) (Fig. 1B). In all cases, coupling to BioID1 included a short flexible linker (GGGGGGKGA) and a FLAG epitope organised with respect to the protein of interest (POI) as follows; POI-Linker-FLAG-BioID1 or FLAG-BioID1-Linker-POI.

To express these constructs in HeLa cells, we adopted a two-step approach. We initially used CRISPR/Cas9 KO to generate individual clonal HeLa lines for all gene alleles. Into the relevant lines we used lentiviruses to transduce the corresponding BioID1-tagged chimera, and by means of viral titration and puromycin selection, we isolated populations expressing as near to endogenous levels of the chimera as possible (Fig. 1C). We extensively validated the localisation and function of each resulting cell population. Endosomal localisation was confirmed by immuno-fluorescence using specific antibodies, or where suitable antibodies were not available, through anti-FLAG labelling (Supplementary Fig. 1). Functionality was defined through the ability of each chimera to rescue well-established missorting phenotypes of endosomal cargo proteins observed upon KO of SNX27-Retromer and SNX17-Retriever complexes[10,19]. All chimeric versions of VPS35 and SNX27 rescued the lysosomal missorting of the GLUT1 glucose transporter and promoted its recycling to the cell surface (Supplementary Fig. 2A, B). Similarly, SNX17 and the VPS35L chimeras rescued the lysosomal missorting of α5β1-integrin and promoted its cell surface recycling (Supplementary Fig. 2C, D). The chimeric version of HRS was able to rescue the loss of expression of the other component of the ESCRT-0 complex, STAM1/STAM2 and rescued the morphology of the EEA1 and SNX1-positive endosomes, consistent with its integration into a functional complex (Supplementary Fig. 2E–G).

To explore the robustness of the proximity labelling, we performed an initial biased identification of the biotinylated proteins using western analysis of target proteins. Although 12 h incubation with biotin was enough to label some interactors, we chose to perform all experiments using 24 h incubation to maximise the detection of weak and low abundance proximity proteins, and the potential to identify cargo proteins transiting through the sub-domains (Supplementary Fig. 2H). Importantly, we tested whether the actual biotin-labelling process had any detectable effect on Retromer and Retriever-dependent cargo sorting. After 24 h of incubation with 50 μM biotin, we did not detect significant changes in the trafficking of GLUT1 or α5β1-integrin in any of the HeLa lines expressing the chimeric versions (Supplementary Fig. 2A–D). This suggests that the BioID1 protocol does not induce a detectable perturbation in endosomal sorting through a build-up of inhibitory biotin on the functional machinery within the sorting sub-domains.

Across the different BioID1 chimeric HeLa cell lines the level of each biotinylated protein was comparable when blotted using the same FLAG antibody, being heterogeneous but always lower than the cytosolic BioID1 (Fig. 1C). Parallel comparison of the biotinylated proteins after 24 h incubation established minimal labelling of SNX27-Retromer, SNX17-Retriever, and HRS in parental HeLa cells lacking any BioID1 expression, and HeLa cells expressing cytoplasmic BioID1 or TOM70-RFP-BioID1 (Fig. 1D). For Retromer, BioID1-VPS35 and VPS35-BioID1 revealed strong labelling of endogenous VPS26 and FAM21, a component of the WASH complex that directly associates with VPS35[24,42–44]. Labelling was also detected for SNX1, a component of the ESCPE-1 complex[27], but unexpectedly not SNX27[10,11]. For Retriever, VPS35L-BioID1 and BioID1-VPS35L labelled endogenous VPS26C and FAM21, consistent with the known indirect association of Retriever with the WASH complex[19]. BioID1-SNX17 showed limited labelling for the blotted proteins. In contrast, BioID1-SNX27 showed robust labelling of FAM21 and SNX1 entirely consistent with its known direct interaction with these proteins[10,29–31,44]. Finally, BioID1-HRS labelled endogenous STAM1 with minimal detectable binding to any proteins from Retromer, Retriever, and the ESCPE-1 and WASH complexes. These data provide proof-of-principle that the experimental design is

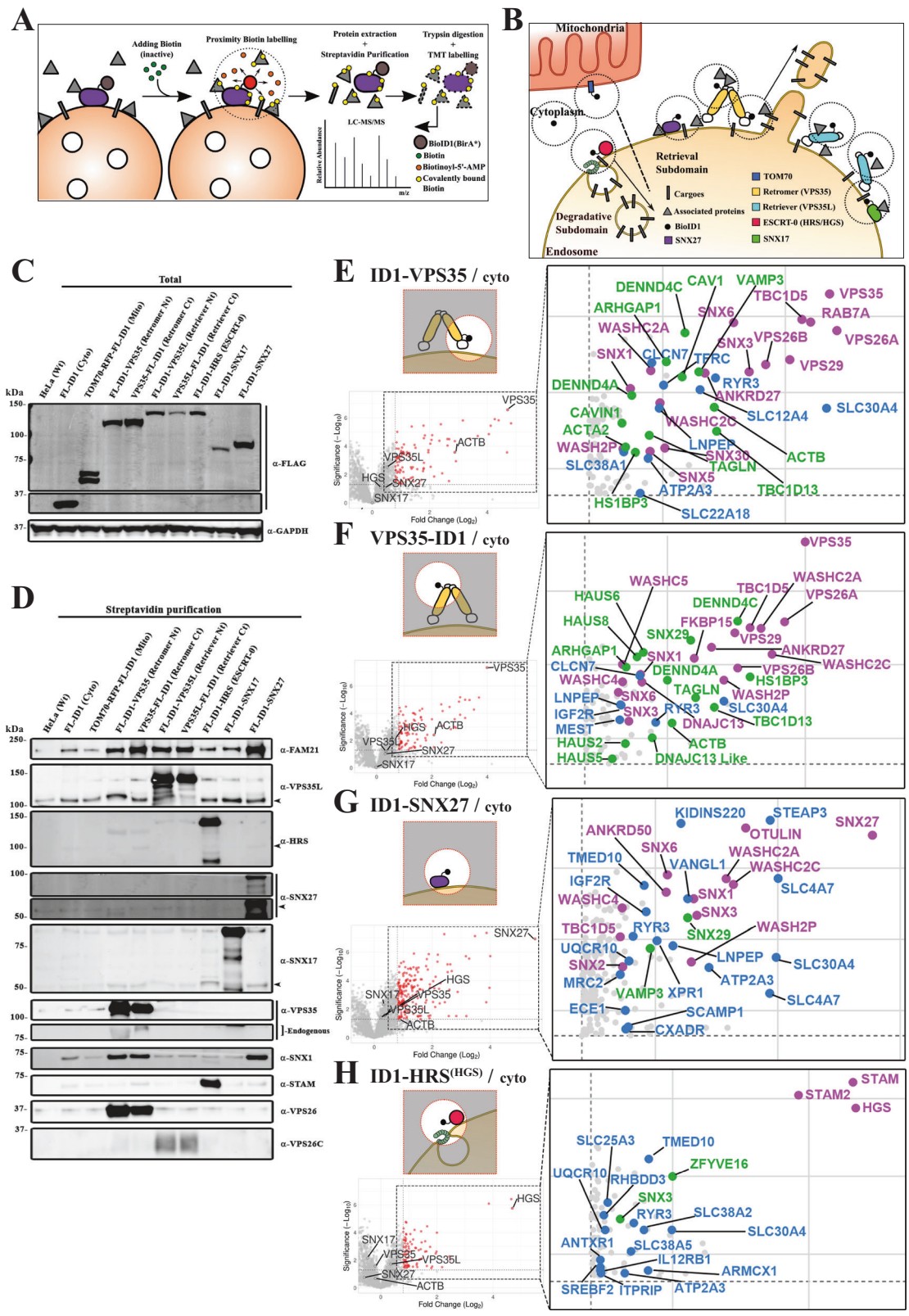

suitable for providing biochemical insight into the organisation of the endosomal sorting sub-domains.

## Unbiased quantitative proximity proteomics of endosomal sorting sub-domains

To unbiasedly identify the proximity proteomes, we performed 10-plex TMT-based quantitative proteomics across 6 independent experimental repeats comparing the protein abundances between BioID1-VPS35, VPS35-BioID1, BioID1-VPS35L, VPS35L-BioID1, BioID1-SNX17, BioID1-SNX27, BioID1-HRS and the parental, cytoplasmic BioID1 and TOM70-RFP-BioID1 controls. The resulting data were compared between the individual endosomal BioID1 chimera and the cytosolic BioID1 in volcano plots of the Fold Change (Log$_2$) versus statistical significance (-Log$_{10}$) (Figs. 1E–H and 2A–C).

**Fig. 1 | Proximity proteomics of endosomal sorting sub-domains. A** Schematic representation of proximity-dependent biotin identification (BioID) of a protein of interest (POI) over the endosomal surface. **B** Cartoon depicting endosomal proteins tagged with BioID1, including a soluble version called cytosolic BioID1 and a TOM70-directed cytoplasmic facing mitochondrial external membrane. **C** Single experiment western blot showing comparison of total protein levels of the BioID1-tagged proteins expressed in the engineered cell lines using an antibody against the FLAG epitope. **D** Single experiment western blot comparing biotinylated proteins after 24 h biotin incubation, lysis and streptavidin purification among the different cell lines. **E**–**H** Volcano plots representing the proteins detected in proximity to the indicated endosomal proteins when compared with the cytosolic BioID1. $n = 6$ independent experiments, standard two-sided t-test analysis. *X*-axis denotes Fold Change (FC) in $\log_2$ scale and *Y*-axis statistical significance in $-\log_{10}$ scale format; for reference, 1.3 $-\log_{10}$ is equal to 0.05 *p*-value. The right insets are magnifications showing only significant proteins ($\geq 1.3$) with FC $\geq 0.8$. In magenta, already known interactors, in blue, transmembrane proteins, in green, unrecognised or barely studied interactors. **E** BioID1-VPS35. **F** VPS35-BioID1. **G** BioID1-SNX27. **H** BioID1-HRS/HGS.

As expected, both BioID1-VPS35 and VPS35-BioID1 labelled the other Retromer subunits VPS29, VPS26A and VPS26B and several previously identified Retromer accessory proteins including SNX3, RAB7A, TBC1D5, ANKRD27, DENND4A and DENND4C, FKBP15, and the WASHC2A (FAM21A/B), WASHC2C (FAM21C), WASHC5 (strumpellin), WASHC4 (SWIP) and WASH2P (WASH) subunits of the WASH complex[8,24,42,43,45–49] (Fig. 1E, F).

For BioID1-VPS35L and VPS35L-BioID1, the Retriever subunit VPS26C was identified along with the coiled coil proteins CCDC22 and CCDC93, which form the backbone of the CCC complex that associates with Retriever to form, alongside DENND10, the Commander super-assembly[17,19,21,22] (Fig. 2A, B). The identification of WASHC2C (FAM21C), WASHC2A (FAM21A/B), WASHC3 (SWIP) and WASH2P (WASH) is consistent with the known association of the CCC complex with the WASH complex[17] (Fig. 2A, B). Finally, ID1-HRS/HGS showed robust labelling of the other ESCRT-0 subunits STAM1 and STAM2, but there was no significant labelling of Retromer and associated accessory proteins (the exception being SNX3), Retriever or the CCC and WASH complexes (Fig. 1H). Together, these data provide an unbiased validation of the experimental design and reveal little proximity overlap between the molecular components of the ESCRT-0 demarcated degradative sub-domain and the retrieval sub-domain defined by VPS35 and VPS35L.

For the Retromer cargo adaptor SNX27, ID1-SNX27 labelled a wide range of known interactions, including the WASH subunits WASHC2A (FAM21A/B), WASHC2C (FAM21C), WASHC4 (SWIP) and WASH2P (WASH), the ESCPE-1 subunits SNX1, SNX2 and SNX6[27,29–31,44], ANKRD50[50] and Otulin[51] (Fig. 1G). Multiple transmembrane proteins were also labelled, many of which contain PDZ binding motifs and rely on SNX27 for their endosomal retrieval and recycling; examples included KIDINS220, STEAP3, SLC4A7 and VANGL1[10,12] (Fig. 1G). While BioID1-SNX27 labelled the core Retromer subunit VPS35, it failed to reach the cut-off for $\log_2$ fold change ($\geq 0.8$) and significance (Fig. 1G), and similarly, BioID1-VPS35 or VPS35-BioID1, while labelling SNX27, failed to reach the same cut-off parameters (Fig. 1E, F). Given the direct nature of the PDZ domain of SNX27 in binding to VPS26A and VPS26B and the importance of this coupling for Retromer-mediated cargo sorting[11], the lack of significantly enriched labelling may result from a technical limitation or reflect a highly dynamic, transient association between cargo-loaded SNX27 and the Retromer demarcated sub-domain during the handover of captured cargo[31]. BioID1-SNX17, on the other hand, labelled the VPS26C subunit of Retriever consistent with the mechanism of SNX17 binding to this heterotrimer[19,52,53] (labelling of VPS35L fell just below the $\log_2$ fold change cut-off) (Fig. 2C).

Unbiased analysis of all the significant protein hits with FC $\geq 0.8$ ($\log_2$) using metascape.org platform[54], revealed top enriched ontology terms for categories including Cellular Component, Biological Processes and CORUM (Supplementary Fig. 3). With the expected exception of Mito-ID1, all protein lists showed significant enrichment for the GO term endosomal transport and additional terms including 'endocytic recycling', 'regulation of transport', 'Retromer complex, Retromer complex binding and Retromer tubulation complex', 'CCDC22-CCDC93-C16orf62-FAM21-WASH complex', 'WASH complex', and 'regulation of actin filament-based process'. From analysis of Disease Processes, the data set revealed a significant enrichment of the

GO terms associated with a range of neurological conditions, metabolic syndromes and cancers consistent with the emerging functional role of the endosomal retrieval sub-domain in neuroprotection, metabolic regulation and growth factor signalling (Fig. 2D and Supplementary Fig. 3). Overall, these data further validate the experimental design and reveal the global strength of the acquired proximity proteomes.

## Quantitative comparison among endosomal sub-domains

As our 10-Plex TMT-based experimental design allowed all proximity proteomes to be collected within the same proteomic run, we used the platform prohits-viz.org[55] and specifically the dot plot option to better visualise the quantitative enrichment and statistical significance of each identified protein across all data sets (Supplementary Fig. 4A). For the Retromer VPS26A/B:VPS35:VPS29 and Retriever VPS26C:VPS35L:VPS29 complexes this allowed the clear visualisation that ID1-VPS35 and VPS35-ID1 displayed a significant enrichment of VPS26A, VPS26B and VPS29, an enrichment that was not detected through BioID1 tagging of VPS35L, HRS (ESCRT-0 subunit) or the MITO-ID1 (Fig. 2E). The same was also true for VPS35L-ID1 and ID1-VPS35L where VPS26C was clearly detected but only moderately detected by ID1-VPS35. The lack of VPS29 enrichment in either VPS35L proximity proteomes may reflect a technical caveat or arise from a restricted capacity for biotinylation due to the distinct mechanism for VPS29 association within the Retriever assembly[20–22]. These data point to a level of spatial segregation between Retromer and Retriever within the retrieval sub-domain that, as far as we are aware, has not been resolved by imaging. That said, the membrane distal VPS35-ID1 but not the membrane proximal ID1-VPS35 (see below for more detailed discussion) catalysed labelling of specific subunits of the Retriever-associated CCC complex, namely CCDC22 and CCDC93, suggestive of a proximity between Retromer and Retriever containing Commander super-assembly. Building on this, the common labelling of the WASH complex by both Retromer and Retriever (Fig. 2E) is entirely consistent with their known biochemical association and physical co-localisation to the retrieval sub-domain and the central role of actin polymerisation and actin turnover in Retromer- and Retriever-dependent cargo sorting[17,19,24,56–58]. Indeed, β-actin was detected by Retromer and Retriever BioID1s, but not in BioID1-HRS, consistent with the known segregation of branched filamentous actin with the retrieval sub-domain (Supplementary Fig. 4A).

## Proximity proteomics provides biochemical insight into Retromer's organisation in the retrieval sub-domain

Cryo-ET analysis of membrane-associated Retromer has established a polarised organisation where the amino-terminal region of VPS35 associates with VPS26A/B proximal to the membrane surface and the membrane distal carboxy-terminal region of VPS35 dimerises with a neighbouring Retromer to form an arch-like structure (Fig. 2F and Supplementary Fig. 4B)[13,14,16]. By including two BioID1 versions of VPS35, we sought to provide *in cellulo* biochemical insight into the organisation of Retromer relative to the endosomal surface, reasoning that the biotin ligase in ID1-VPS35 should be relatively close to the endosomal membrane (Fig. 2F, **upper panel**), while in the VPS35-ID1 version it should be towards the top of the arches, more distal to the

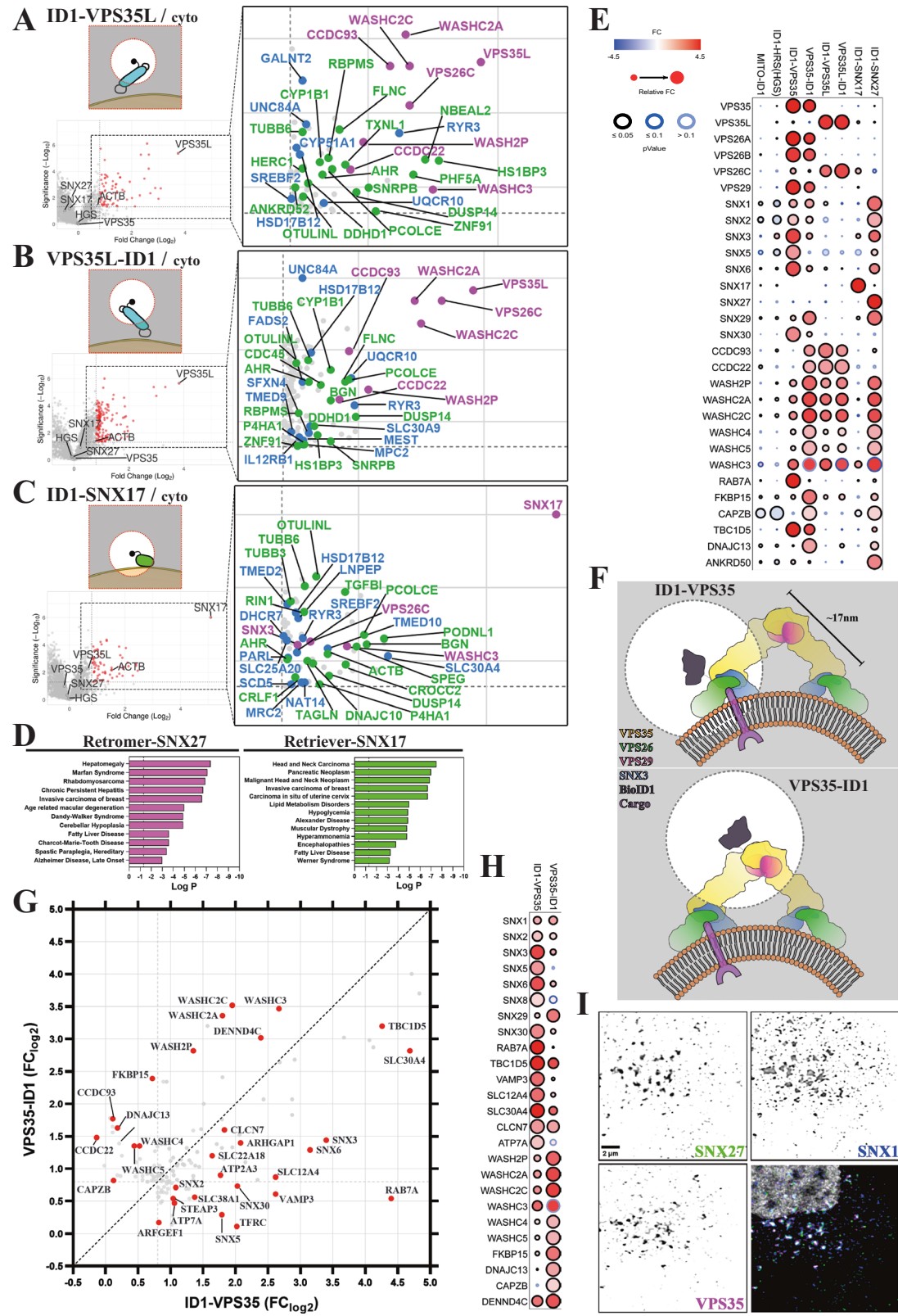

membrane (Fig. 2F, **lower panel**). For those identified proteins with FC ≥ 0.8, we plotted ID1-VPS35 versus VPS35-ID1 to observe any quantitative difference in labelling efficiency across the proximity proteomes (Fig. 2G). This revealed that the membrane proximal ID1-VPS35 preferentially labelled several transmembrane proteins, including well-established Retromer cargo proteins such as STEAP3, TFRC and ATP7A (Fig. 2G)[10]. At the mechanistic level, ID1-VPS35 preferentially labelled

the membrane anchored RAB7 GTPase, which plays a central role in the endosomal recruitment of Retromer through binding to VPS35[45,46], and the phosphatidylinositol 3-monophosphate (PI(3)P) binding sorting nexin SNX3, which by associating with the VPS26A/B:VPS35 interface assists in the endosomal association of Retromer and the sequence-dependent recognition of cargo undergoing SNX3-Retromer mediated endosomal sorting (Fig. 2G)[8,13,15,16,59,60]. Additional membrane binding

**Fig. 2 | Proximity proteomes of Retriever (VPS35L) and SNX17, and multiple comparisons among all proximity proteomes.** **A**–**C** Volcano plots representing the proteins detected in proximity to the indicated endosomal proteins (see Fig. 1A) when compared with the cytosolic BioID1. $n = 6$ independent experiments, standard two-sided $t$ test analysis. $X$-axis denotes Fold Change (FC) in $Log_2$ scale and $Y$-axis statistical significance in $-Log_{10}$ scale format; for reference, 1.3 $-Log_{10}$ is equal to 0.05 $p$-value. The right insets are magnifications showing only significant proteins with FC ≥ 0.8. In magenta, already known interactors, in blue, transmembrane proteins, in green, unrecognised or barely studied interactors. **A** BioID1-VPS35L. **B** VPS35L-BioID1. **C** BioID1-SNX17. **D** Retromer-SNX27 (left panel) and Retriever-SNX17 (right panel) proximity proteomes show enrichment for multiple disease-associated GO terms. Significant proteins ($p$-value < 0.05) with FC ≥ 0.8 detected in proximity to SNX27-Retromer or SNX17-Retriever from Fig. 1 and Fig. 2A–C, were combined and analysed with the Metascape webapp to represent enrichment of DisGeneNET terms. Hypergeometric test and Benjamini-Hochberg correction, category-enrichment $p$-value is represented in $-Log10$ scale. Extended version in Supplementary Fig. 3B, C. **E** Dot plot showing the comparison of FC ($Log_2$) and significance of selected endosomal proteins from the experiment in Fig. 1 and

Fig. 2A–C. Proteins shown are significant ($p$-value < 0.05) and with FC ≥ 0.8 in at least one BioID1 comparison with the cytosolic BioID1. Filling colours represent the FC value, edge colours the $p$-value, and the size symbolises the relative FC. Extended version in Supplementary Fig. 4A. **F** Cartoon depicting the expected localisation of the BioID1 biotin ligase, in black, along the arch-like conformation of Retromer over the endosomal surface. Edge-dashed circles represent the 10 nm-radius spheres where BioID1 can biotinylate proteins. Sizes are based on the Retromer arch structure PDB:7BLN. **G** Scatter plot showing the FC values ($Log_2$) for detected proteins in both Retromer proximity proteomes from Fig. 1; FC from BioID1-VPS35 in $X$-axis and FC from VPS35-BioID1 in $Y$-axis. The farther from the central dashed line the protein is, the greater the preferential proximity labelling to a specific BioID1 version. **H** Dot plot showing the comparison between both Retromer BioID1 versions using the same parameters as in (**E**). **I** Colocalization of endogenous Retromer (VPS35), SNX27 and SNX1 in HeLa cells using confocal imaging with adaptive deconvolution to improve lateral resolution. Scale bar – 2 μm. Details in Materials and Methods section and Supplementary Fig. 4D. The experiment was repeated independently twice with similar results.

sorting nexins of the ESCPE-1 complex[27,61], namely SNX1, SNX2, SNX5 and SNX6, were also preferentially labelled by ID1-VPS35 (Fig. 2G, H). In addition, and consistent with the colocalization of endogenous VPS35, SNX27 and SNX1 (Fig. 2I and Supplementary Fig. 4D), ID1-SNX27 also labelled ESCPE-1 subunits (Fig. 2E), supporting the recently proposed cargo handover model for SNX27-Retromer mediated ESCPE-1-dependent cargo sorting[31]. Here, the SNX1 and SNX2 subunits of ESCPE-1 bind directly to the FERM domain of SNX27 to couple sequence-dependent cargo recognition with the biogenesis of tubular transport carriers for the promotion of plasma membrane recycling[29–31]. Indeed, the relative FC of the ID1-SNX27 labelling of SNX1 and SNX2 relative to SNX5 and SNX6 is entirely consistent with this mechanism of coupling (Fig. 2E).

In contrast to the labelling of membrane proximal proteins by ID1-VPS35, all subunits of the WASH complex were preferentially detected by the membrane distal VPS35-ID1 (Fig. 2G, H). This is entirely consistent with the mechanism of WASH complex association, where acidic-Asp-Leu-Phe (aDLF) motifs in the FAM21 subunit bind to VPS29 and two sites towards the carboxy-terminal membrane-distal region of VPS35[44]. Two additional regulators of endosomal actin dynamics the FK506-binding protein-15 (FKBP15)[42,62,63] and the actin binding and capping proteins CAPZA2 and CAPZB[64,65] were also preferentially labelled by VPS35-ID1 (Fig. 2G, H), as was DNAJC13 (also known as RME-8) a protein that by catalysing the removal of encroaching HRS from the degradative sub-domain serves to regulate the separation of degradative and retrieval sub-domains (Fig. 2G, H)[37,66–69]. Together, these data establish biochemical signatures for the inner membrane proximal and outer membrane distal layers of key accessory proteins of the Retromer coat complex.

In performing a similar analysis with ID1-VPS35L and VPS35L-ID1, we failed to detect such clear segregation in the proximity profiles of Retriever accessory proteins (Fig. 2E and Supplementary Fig. 4A). One likely reason is that Retriever (and the larger Commander assembly) may adopt a completely different membrane-proximal orientation to Retromer. It may also reflect the distinct organisation of the Retriever heterotrimer where the amino- and carboxy-termini are less spatially segregated due to its more compact structure[20–22] (Supplementary Fig. 4C). Although VPS35L BioID1 constructs robustly labelled CCDC22 and CCDC93, we did not detect significant enrichment of any COMMD proteins that form the core decameric ring of the CCC complex. This provides *in cellulo* evidence in support of the proposed structural organisation of the Commander super-assembly[20–22].

### Retromer is a hub for GTPase regulation
Having established the robustness of the experimental design and methodology by focusing on known features of Retromer and

Retriever, we next probed for molecular insight into retrieval sub-domain organisation. Taking those proteins identified across all proximity proteins (396 proteins with FC ≥ 0.8 ($log_2$)) we manually sub-classified the data based on functional terms (Fig. 3A). Within these clusters we searched for proteins whose enrichment and statistical significance profiles displayed a heavy bias towards detection by VPS35-ID1 and/or ID1-VPS35 (proteins highlighted by red arrows in Fig. 3A). This identified several potential components of the Retromer sub-domain including the endosome-to-TGN SNARE VAMP3, sorting nexin-29 (SNX29), the caveolae associated caveolin-1 (CAV1) and CAVIN1, subunits of the octameric augmin complex (HAUS2, HAUS5, HAUS6, HAUS8), which interacts with the γ-tubulin ring complex (γTuRC) to stabilise pre-existing microtubules and facilitate microtubule branching during chromosomal segregation and neuronal migration, development, and polarisation[70–75], and the late endosomal $2Cl^-/H^+$ exchanger CLCN7 – dysfunction in this exchanger leads to a lysosomal storage disease and neurodegeneration in humans[76]. VPS35-BioID1 and BioID1-VPS35 proximity labelling, followed by targeted western analysis, confirmed the selective labelling of VAMP3, CAV1 and CAVIN1, and subunits of the augmin complex within the Retromer sub-domain, suggestive of these proteins and complexes associating with and/or transiently traversing through this sub-domain (Fig. 3B, C).

Also present in the Retromer sub-domain were a variety of GTPase-activating proteins (GAPs) and guanine nucleotide exchange factors (GEFs). These included known Retromer accessory proteins, the RAB7 GAP TBC1D5[46,77–81], the RAB21/RAB32/RAB38 GEF VARP (*a.k.a.* ANKRD27)[47,48,82], and the RAB10 GEFs DENND4A and DENND4C[83,84]. Activation of RAB10 is heavily implicated in general endosomal cargo recycling[85–93] and one of its effector proteins TBC1D13, a functional RAB35 GAP was also labelled within the Retromer proximity proteome[94] (Fig. 3A). RAB35 itself regulates endosomal exit for cell surface recycling[95–99], in part through controlling localised actin dynamics and phosphoinositide metabolism[100,101]. Also labelled within the Retromer proximity proteome was the Cdc42 GAP ARHGAP1 (*a.k.a.* cdc42GAP), and two proteins selectively labelled by ID1-VPS35 over VPS35-ID1, the Ras GAP neurofibromin (NF1) and ARFGEF1 (*a.k.a.* BIG1), a GEF linked with regulation of the Golgi localised ARF1 and ARF3 GTPases (Fig. 3A).

VPS35 BioID1 proximity labelling, followed by targeted western analysis confirmed the selectivity of labelling for these proteins within the Retromer sub-domain (Fig. 3B, C). To probe for the potential direct association of these GEFs and GAPs with Retromer, we turned to AlphaFold2 modelling[102–104]. Consistent with published structural data, AlphaFold2 predicted the association of a conserved surface-exposed hydrophobic cavity on VPS29 (defined by Leu152) with Pro-Leu (PL) motifs from TBC1D5 (141PL142) and VARP (713PL714) and for TBC1D5, the

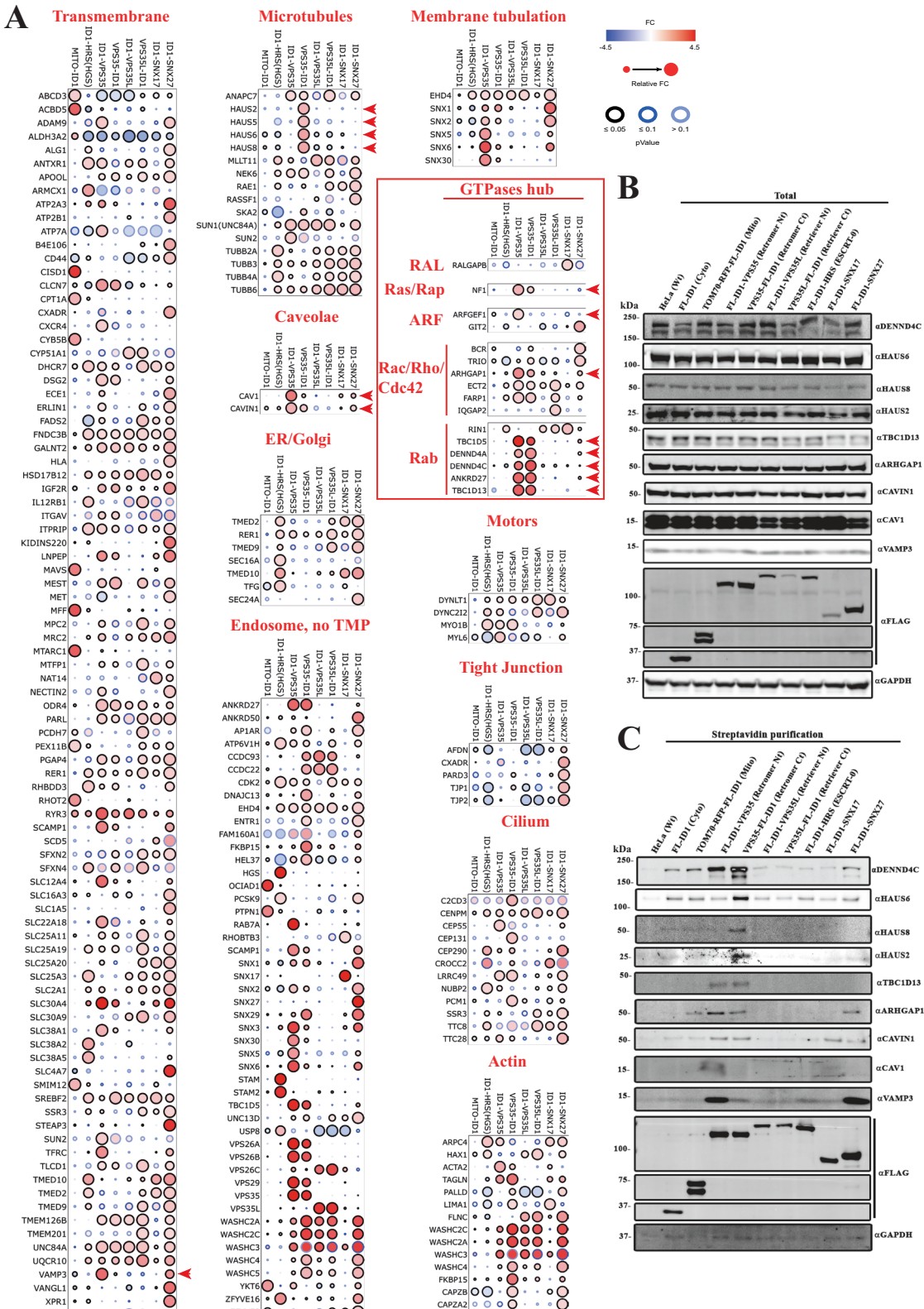

**Fig. 3 | Functional clustering comparing the proximity proteomes. A** Dot plot of proteins detected from the experiment in Figs. 1 and 2A–C, organised by function, pathway or topology. Proteins shown are significant (*p*-value < 0.05) and with FC ≥ 0.8 in at least one BioID1 comparison with the cytosolic BioID1. Filling colours represent the FC value, edge colours the *p*-value, and the size symbolises the relative FC. Red arrowheads highlight proteins displaying a heavy bias towards detection by VPS35-ID1 and/or ID1-VPS35. **B** Single experiment comparison of total protein levels among all the BioID1 cell lines using antibodies for potential Retromer interactions, and (**C**) of the biotinylated proteins after 24 h biotin incubation, lysis and streptavidin purification among the different cell lines using antibodies targeting Retromer proximity proteins.

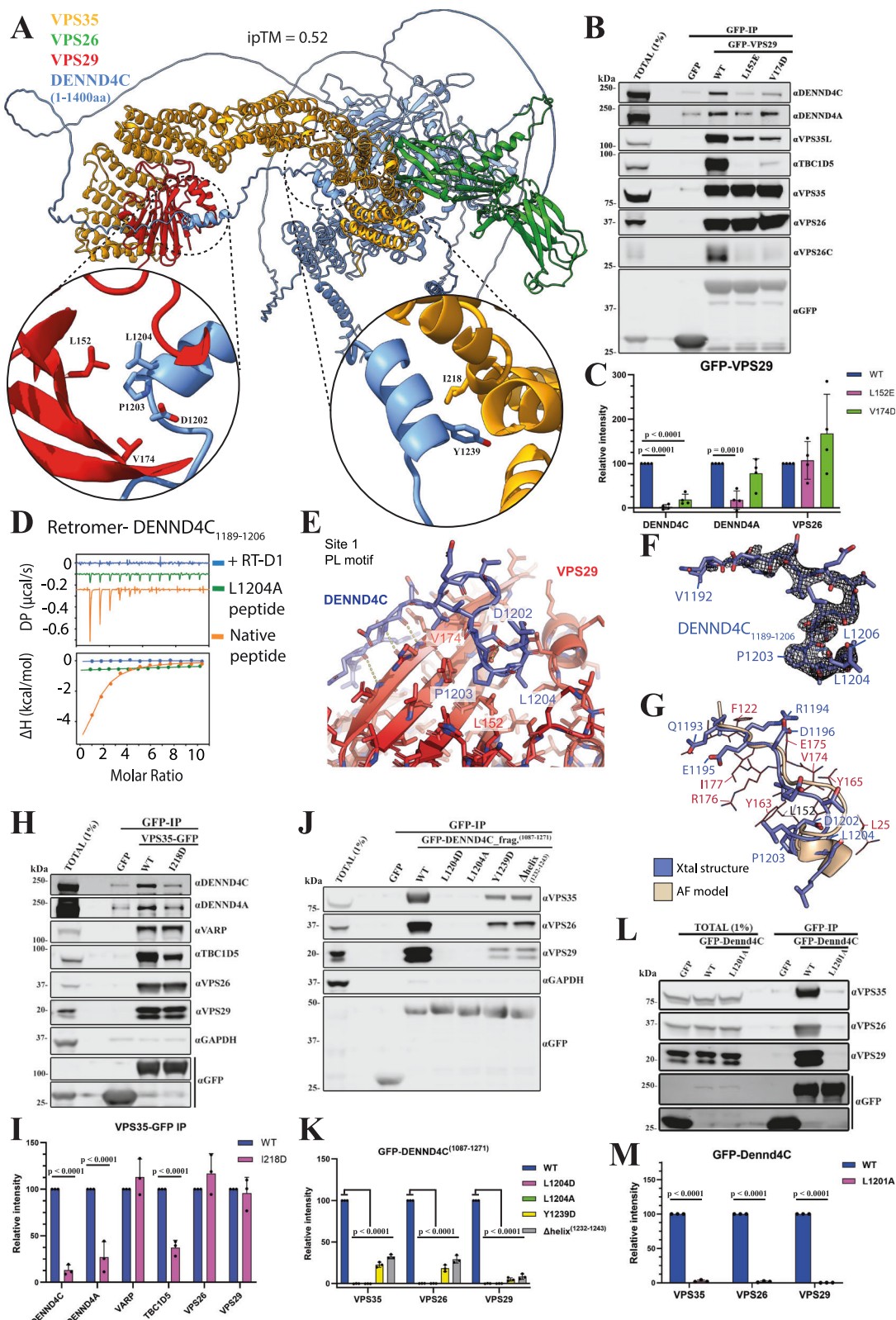

association of an additional [283]Ile-Pro-Phe[285] hydrophobic motif with VPS35 (Supplementary Fig. 5)[77,82]. Highly confident models were derived for Retromer association with DENND4A and DENND4C (Supplementary Fig. 6).

The three DENND4 homologues in humans, DENND4A, 4B and 4 C, all have similar domain structures consisting of an amino-terminal MABP domain (MVB12-associated β-prism domain), DENN domain

(consisting of the three structural modules uDENN, cDENN and dDENN), a long extended unstructured linker sequence, and a carboxy-terminal domain that has a mixed α/β globular structure of unknown function (Supplementary Fig. 7). The carboxy-terminal domain is predicted to fold into a globular structure that forms a tight intra-molecular association with the amino-terminal MABP-DENN domain module. Searches using Foldseek[105] suggest that it is a unique structure

**Fig. 4 | Molecular details of DENND4A and DENND4C binding to Retromer.**
**A** AlphaFold2 prediction of DENND4C (1-1400 aa) interacting with Retromer (rank 1 prediction). Left inset, primary binding site consisting of a conserved PL motif (DENND4C [1203]PL[1204]) interacting with the Leu152 containing hydrophobic cavity of VPS29. Right inset, second binding site predicted between a short DENND4C α-helical stretch (residues 1232–1243) with α-helices towards the amino-termini of VPS35. **B, C** GFP-based co-immunoprecipitation (co-IP) and quantification of GFP-VPS29 wild-type (WT) and hydrophobic pocket mutants after transient transfection in HEK293T cells. Quantitation and statistical analysis of relative band intensity for the indicated proteins, normalised to GFP band intensity. $n = 3$ independent experiments. 1-way ANOVA with Dunnett's multiple comparison test, data presented as mean values relative to WT and error bars represent standard deviation (SD). **D** ITC of DENND4C peptides (1189–1206 aa) binding to Retromer in the absence and presence of competing cyclic peptide RT-D1. **E** Crystal structure of VPS29 in complex with the DENND4C PL motif-containing peptide[1189-1206] (PDB:8VOD). **F** Ribbon representation of the DENND4C PL-motif containing peptide[1189-1206]. The electron density shown corresponds to a simulated-annealing OMIT Fo - Fc map contoured at 3σ. **G** Overlay of the DENND4C peptide structures (slate blue) derived from the crystal structure and from AlphaFold2 prediction (wheat). **H, I** GFP-based co-IP of VPS35-GFP WT and I218D mutant after transient transfection in HEK293T cells. Quantitation and statistical analysis of relative band intensity normalised to GFP band intensity. $n = 3$ independent experiments. Two-sided t-test analysis, data presented as mean values relative to WT and error bars represent SD. **J, K** GFP-based co-IP of GFP-DENND4C fragment[1087-1271], WT and mutants after transient transfection in HEK293T cells. Quantitation and statistical analysis of relative band intensity for the indicated proteins, normalised to GFP band intensity. $n = 3$ independent experiments. 1-way ANOVA with Dunnett's multiple comparison test, data presented as mean values relative to WT and error bars represent SD. **L, M** GFP-based co-IP after cell lysis of HEK293T cells stably expressing GFP or GFP-Dennd4C full-length from mouse, WT and L1201A mutant; equivalent to human L1204A. Quantitation and statistical analysis of relative band intensity for the indicated proteins, normalised to GFP band intensity. $n = 3$ independent experiments. Two-sided $t$ test analysis, data presented as mean values relative to WT and error bars represent SD. Only changes with $p < 0.05$ are shown.

**Table 1 | Thermodynamic parameters for the binding of Retromer and DENND4C by ITC**

| | $K_d$ (μM) | $\Delta H$ (kcal/mol) | $\Delta G$ (kcal/mol) | $-T\Delta S$ (kcal/mol) |
|---|---|---|---|---|
| hRetromer | | | | |
| hDENND4C[1189-1206] | 15.4 ± 0.85 | −10.1 ± 1.41 | −6.6 ± 0.04 | −3.5 ± 1.44 |
| hDENND4C[1189-1206] L1204A | No binding detected | | | |
| hDENND4C[1189-1206] + RT-D1 | No binding detected | | | |

found only in the DENND4 family members. Both DENND4A and DENND4C are predicted to interact with Retromer using identical mechanisms, and we discuss DENND4C for simplicity. AlphaFold2 modelling predicts that a conserved PL motif ([1203]PL[1204]) associates with the Leu152 containing hydrophobic cavity of VPS29 (Fig. 4A). Consistent with these models, GFP-nanotrap immuno-isolation of GFP-VPS29 revealed clear association with endogenous DENND4C, an association that like TBC1D5 was largely dependent on the conserved surface-exposed hydrophobic cavity of VPS29 as shown by VPS29 mutations L152E and V174D (Fig. 4B, C). ITC analysis of a PL-containing synthetic peptide from DENND4C established direct binding to recombinant VPS29 with an affinity ($K_d$) of 15.4 μM, that was blocked by mutation of the PL motif or by the addition of a competing VPS29-binding macrocyclic peptide RT-D1 (Fig. 4D, and Table 1)[106]. We determined the crystal structure of VPS29 in complex with the DENND4C [1203]PL[1204]-containing peptide which completely validated the mode of association (Fig. 4E–G). Furthermore, the crystal structure confirmed the prediction that DENND4C engages VPS29 through both the core PL motif as well as an extended upstream sequence that forms a β-strand extension to the VPS29 structure. AlphaFold2 analysis also predicted a similar mechanism of binding to DENND4A that we confirmed by immuno-isolation (Fig. 4B, C). Despite their presence in the BioID proximity proteomes, no association was detected for ARHGAP1, ARFGEF1 or NF1 in co-immunoprecipitation and Western blot analyses (Supplementary Fig. 6H).

While the observed binding of DENND4C to VPS29 is consistent with previous related crystal structures, a second binding site is also predicted between a short DENND4C α-helical stretch (residues 1232–1243) with α-helices towards the amino-termini of VPS35 (Fig. 4A and Supplementary Fig. 6). Consistent with this AlphaFold2 model, immuno-isolation of VPS35-GFP showed association with endogenous DENND4C that was partially reduced upon mutation of residue Ile218 at the amino-termini of VPS35 (Fig. 4H and I). This suggests that DENND4C (and DENND4A) engages Retromer at two distinct sites: a primary site on VPS29 and a secondary site towards the amino-terminal region of VPS35. Consistent with this, immuno-isolation of GFP-tagged DENND4C (a fragment encoding residues 1087–1271)

confirmed binding to endogenous Retromer (Fig. 4J, K). This was almost completely lost in mutants targeting the PL motif, Leu1204 (DENND4C(L1204A) and -(L1204D)), and significantly reduced by targeting the secondary site through deletion of the short α-helix (residues 1232–1243) or targeted mutagenesis of Tyr1239 (Fig. 4J, K). ConSurf[107] analysis of amino acid sequence conservation revealed a high degree of evolutionary conservation for both interacting interfaces (Supplementary Fig. 6F).

Confirming the *in cellulo* relevance of the Retromer interaction with DENND4A and DENND4C, the colocalization of endogenous DENND4C with the endosomal marker SNX1 was partially reduced in a VPS35 knock out HeLa cell line (Supplementary Fig. 8A, B), and while full-length GFP-Dennd4C associated with Retromer decorated endosomes this association was lost with a GFP-Dennd4C(L1201A) mutant that lacked the ability to bind to Retromer, this mutant targets the equivalent Leu1204 PL motif residue in human DENND4C (Figs. 4L, M and 5A, B). Double DENND4A and DENND4C knock out in HeLa cells (DENND4A + C KO) revealed a partial defect in the steady-state cell surface levels of GLUT1 (Fig. 5C and Supplementary Fig. 8C). Rescue of the DENND4C + A KO cells with GFP-Dennd4C increased steady-state cell surface levels of GLUT1 while the L1201A mutant do not (Fig. 5C, D). Finally, we lentivirally expressed dominant negative RAB10(T23N) in wild-type HeLa cells. This revealed a reduced steady-state surface level of Retromer cargoes GLUT1, KIDINS220 and the copper-dependent cell surface recycling of CTR1[108] (Fig. 5E, F). Together, these data establish the molecular mechanism for the direct association of DENND4A and DENND4C with Retromer and reveal the importance of this interaction for their association to the Retromer retrieval sub-domain. Evidence points to a functional role for these RAB10 GEFs in potentially regulating RAB10-mediated cargo transport through the Retromer subdomain.

### Interactions of Retromer with the TBC domain family of RAB GAPs

We also identified the RAB10 effector and RAB35 GAP TBC1D13 in both BioID1-VPS35 and VPS35-BioID1 experiments[94] (Fig. 1E, F). We therefore modelled the interaction of TBC1D13 with Retromer using AlphaFold2,

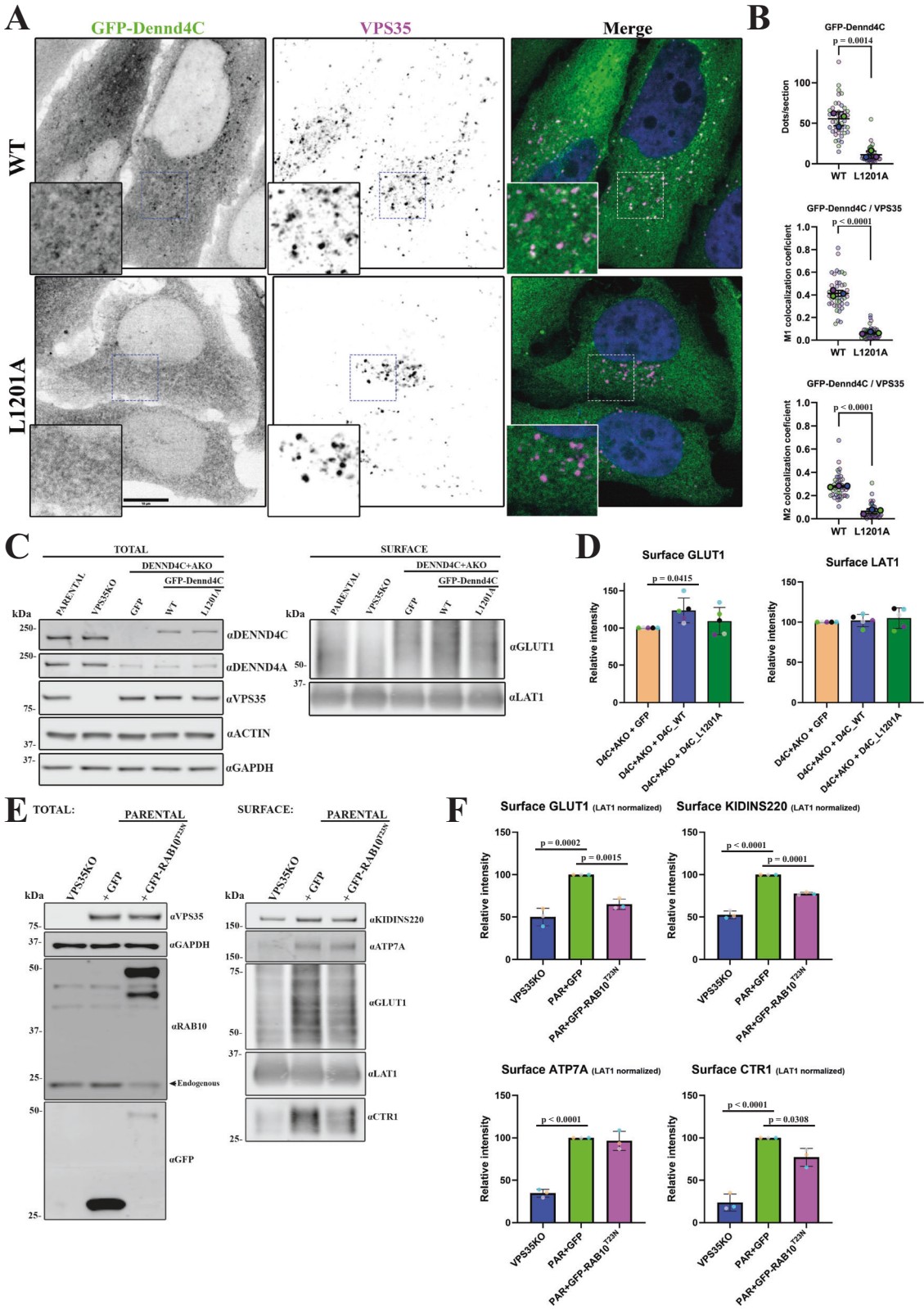

which consistently predicted a high-confidence assembly (Fig. 6A and Supplementary Fig. 9). This revealed a remarkable complex whereby TBC1D13 envelops VPS29 using an extensive clamp structure composed of two loop sequences (residues 80–107 and 164–204). The predicted structure reveals three notable interfaces. The first interface involves loop residues 80–107 containing an HPL motif binding to the canonical surface of VPS29, centred on Leu152 (Fig. 6B **view 1**). This is

consistent with the loss of TBC1D13 interaction with VPS29 mutants L152E and V154D (Fig. 6C, D). We further validated the importance of this binding site by ITC, with recombinant full-length human TBC1D13 binding to the Retromer complex with an affinity of 6.3 μM, which was largely blocked by the competing cyclic peptide RT-D1 (Fig. 6E **and** Table 2). GFP-nanotrap isolation of GFP-tagged TBC1D13 further confirmed binding to Retromer that was lost in the TBC1D13 (P101A)

**Fig. 5 | Role of DENND4A/C in Retromer's cell biology. A** Spinning disk confocal images showing colocalization of stably expressed mouse GFP-Dennd4C with endogenous VPS35 in DENND4C and DENND4A double KO Hela cells; wild-type (WT) and L1201A mutant equivalent to human DENND4C L1204A. Scale bar – 10 μm. **B** Quantitation and statistical analysis of the number of dots per cell section of GFP-Dennd4C and its colocalization with VPS35 from (A) using ImageJ colocalization JACOP plugin, Manders coefficients M1/M2. $n = 3$ independent experiments. Two-sided $t$ test analysis, data presented as mean values relative to WT and error bars represent SD. **C** Analysis of protein surface levels in Hela parental, VPS35KO and DENND4A + C double KO transduced with GFP alone, GFP-Dennd4C WT or L1201A mutant. The VPS35KO cell line was used as a control for Retromer's role in GLUT1 recycling. The left panel shows total protein levels, while the right panel shows the cell surface protein fraction after surface biotinylation and streptavidin

purification. **D** Quantitation and statistical analysis of GLUT1 and LAT1 surface levels in transduced Hela DENND4A + C double KO from experiment (C). $n = 5$ independent experiments. 1-way ANOVA with Dunnett's multiple comparison test, data presented as mean values relative to WT and error bars represent SD. **E** Analysis of protein surface levels in Hela cells transduced with GFP alone or GFP-RAB10 T23N dominant negative mutant. The VPS35KO cell line was used as a control for Retromer's role in protein recycling. The left panel shows total protein levels, while the right panel shows the cell surface protein fraction after surface biotinylation and streptavidin purification. **F** Quantitation and statistical analysis of GLUT1, KIDINS220, ATP7A and CTR1 surface levels in Hela cells from experiment (E). Data was normalised to LAT1. $n = 3$ independent experiments. 1-way ANOVA with Dunnett's multiple comparison test, data presented as mean values relative to WT and error bars represent SD. Only changes with $p < 0.05$ are shown.

mutant that targeted the HPL motif (Fig. 6F, G). On the opposite face of VPS29, a second interface is formed by residues 164-204 of TBC1D13, which form an extended β-hairpin structure and makes a direct contact with the VPS35 subunit at their extremity (Fig. 6B view 2). The third notable prediction is that the amino-terminal methionine of VPS29 is largely buried in the structure (Fig. 6B view 3). This suggests that the full binding affinity for TBC1D13 will require the canonical VPS29 isoform 1, and that the splice variants 2 and 3 with longer amino-terminal sequences[109] could be compromised for TBC1D13 binding. Finally, when expressed in cells, full-length GFP-tagged TBC1D13 localised to Retromer-decorated endosomes, an endosomal targeting that was completely lost with the TBC1D13(P101A) Retromer-binding mutant (Fig. 6H, I).

### In silico screening identifies additional RAB GEFs and RAB GAPs as Retromer accessory proteins

To further explore the ability of Retromer to potentially associate with RAB GEFs and RAB GAPs, we performed an in-silico screen of all human DENND and TBC domain-containing family members. We applied AlphaFold2 and Alphafold3[110] to screen Retromer using an approach similar to our previous studies of the Mint interactome (Figs. 7A, B and Supplementary Fig. 10)[111]. To carry out confident prediction over non-specific associations, Alphafold modelling was carried out using Retromer, a Retromer sub-complex or individual Retromer subunits against TBCs/DENNDs (Fig. 7A, B). The IPTM scores generated by different runs were combined to pick up those that showed consistent binding to specific Retromer subunits with a high IPTM score over non-specific interactors. Using this approach, among the 14 human DENND family members, only DENND4A and DENND4C were predicted to bind VPS35:VPS29 with high confidence (Fig. 7A, B). Further validation using pDOCKQ and SPOC confirmed that both DENND4A and DENND4C associated with the VPS29 subunit (Fig. 7B). The poorly characterised DENND11 also possessed a PL motif but did not display a sufficient confidence binding score with Retromer subunits, and expressed GFP-DENND11 showed weak association with Retromer when immuno-isolated from HEK cells, independently of the PL motif (Supplementary Fig. 10C).

As for TBC domain-containing GAPs, confident prediction was observed for VPS29 with TBC1D13 and the previously characterised TBC1D5[77]. None of the TBCs and DENNDs showed high binding confidence to the VPS26A:VPS35 interface (Fig. 7A). Screening among the TBCs with low confidence in binding to Retromer, VPS29 was predicted to bind PL motifs in both TBC1D1 and TBC1D4, GAPs for RABs that include RAB10[112] (Fig. 7B). Both TBCs reveal binding to the conserved hydrophobic cavity of VPS29 through their PL motif (86DPL88 in TBC1D1 and 14HPL16 in TBC1D4) but with limited interacting residues (Supplementary Fig. 11). Indeed, when expressed as GFP fusions and GFP-trap immuno-isolated GFP-TBC1D1 and GFP-TBC1D4 showed limited binding to endogenous Retromer (Fig. 7C). Further Alphafold2 screening in combination with SPOC and pDOCKQ of the PL motif containing TBCs/DENNDs, revealed that TBC1D1 and TBC1D4 are likely

to form a heterodimer through their C-terminal tails (Supplementary Fig. 12). Similar dimerisation was also observed in TBC1D5, showing a small region within the C-terminal disordered tail likely involved in homodimerization (Supplementary Fig. 12). In TBC1D1 and TBC1D4, the disordered tail region after the TBC domain likely responsible for both homo- and heterodimerization (Supplementary Fig. 12). To examine whether heterodimerization enhances binding to Retromer, we co-expressed GFP-TBC1D1 with mCherry-TBC1D4 and performed GFP-nanotrap immuno-isolation. This established that these RAB GAPs do indeed form heterodimers and that this appears to enhance association with endogenous Retromer (Fig. 7D, E). Under these conditions, mutagenesis of both PL motifs, TBC1D1(P87A) and TBC1D4(P15A), significantly reduces Retromer association (Fig. 7D, E).

Overall, with multiple copies of Retromer being enriched within the retrieval sub-domain and lower concentrations across the limiting endosomal membrane, these data provide mechanistic evidence for a model where Retromer associates with selected RAB GEFs and RAB GAPs to regulate the activity of key endosomal RAB identity cues (Fig. 7F). These include RAB7, RAB10, RAB21, and RAB35 and most likely other RAB GTPases, given the broad spectrum of RAB substrates for some of these regulators.

## Discussion

Central to achieving a thorough understanding of the mechanistic basis and functional significance of endosomal cargo retrieval and recycling will be a detailed appreciation of the components of the retrieval sub-domain, their organisation, and their dynamic regulation[1,113]. Here, we have engineered proximity proteomic technology to provide a quantitative approach for the biochemical identification of the components of the endosomal Retromer and Retriever demarcated retrieval sub-domain. This has revealed quantitative in-cellulo biochemical data supportive of the structure and organisation of Retromer and Retriever and a wealth of molecular insight to pave the way to a greater mechanistic understanding of Retromer and, more generally, retrieval sub-domain structure, organisation and function in human health and disease.

Through designing our BioID strategy into a single proximity proteomic analysis, we have directly compared the quantitative proximity signatures of the HRS-demarcated degradative sub-domain with the Retromer and Retriever defined retrieval sub-domain. This has established a clear biochemical separation in the proximity profile of these degradative and retrieval sub-domains, entirely consistent with imaging data of their spatial separation on the limiting membrane of individual endosomes[34]. When comparing Retromer and Retriever, our data suggests that although these heterotrimeric complexes do not appear at steady-state to reside in proximity to one another, they are connected through shared proximity profiles with the WASH complex. This is entirely consistent with the known direct association of Retromer with the FAM21 subunit of the WASH complex[42-44,114], and the less well-described coupling of Retriever to the WASH complex by means of the CCC complex[58]. It is also consistent with the essential role of the

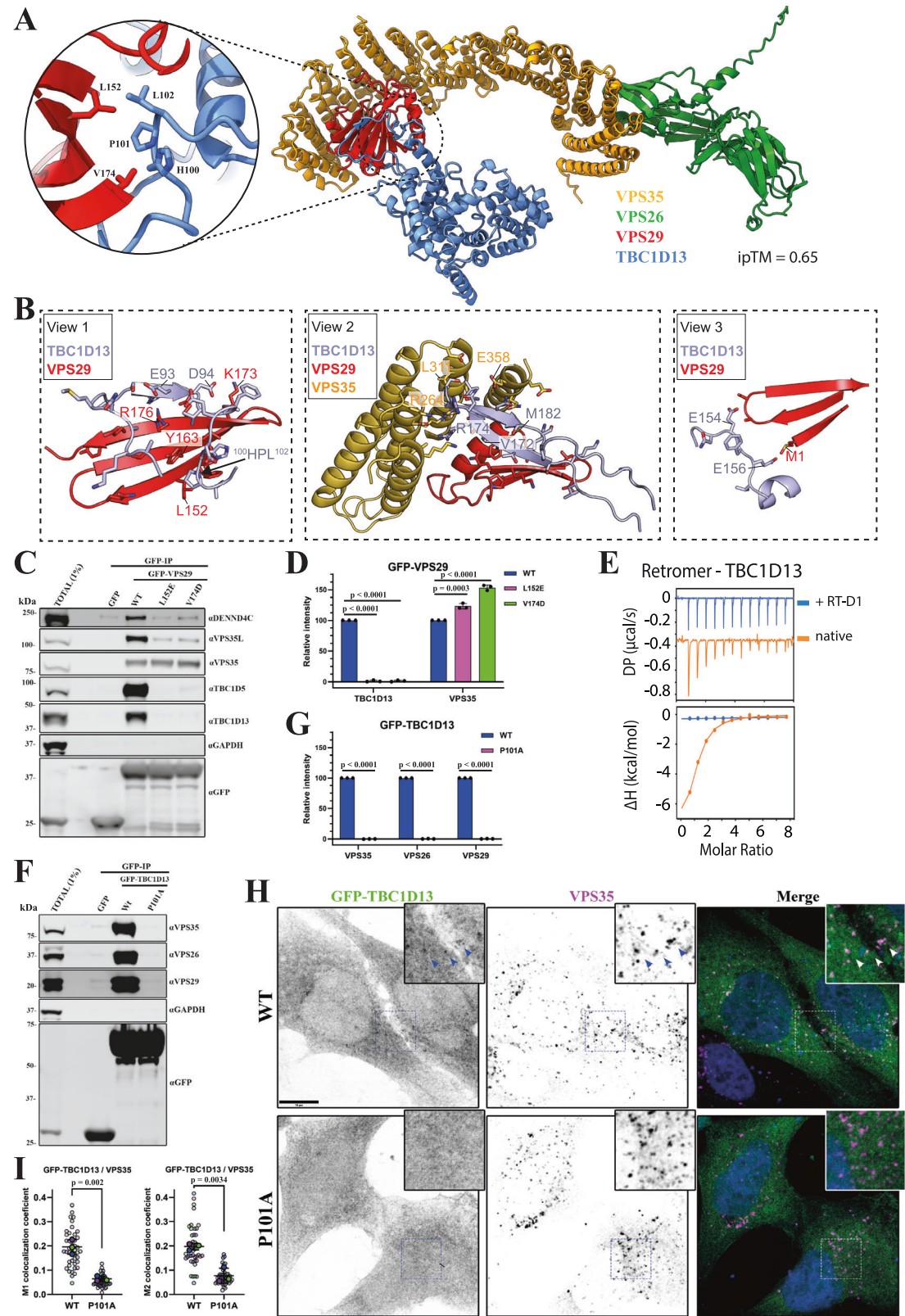

WASH complex and its ability to regulate localised branched F-actin dynamics in the organisation of the Retromer and Retriever demarcated retrieval sub-domain[34,42,43,57,58,115,116].

By strategic engineering of the BioID1 enzyme to the amino-terminus and carboxy-terminus of the extended core VPS35 α-solenoid of Retromer, we have provided *in-cellulo* quantitative biochemical data supporting the arch-like organisation of the membrane-associated

Retromer dimers as observed by cryo-ET of the in vitro reconstituted liposome associated Retromer[14,16]. The proximity profile of the amino-terminal BioID1-tagged VPS35 is consistent with the juxtaposed position of this region to the cytoplasmic-facing endosomal membrane, while the carboxy-terminal BioID1 proximity proteome reflects the membrane distal localisation at the apex of Retromer arches. These data therefore establish biochemical proteinaceous signatures for the

**Fig. 6 | Interactions between TBC1D13 and Retromer complex. A** AlphaFold2 predictive model of TBC1D13 interacting with Retromer (rank 1 prediction). Circular inset shows the main binding site consisting of a conserved PL motif (TBC1D13 [101]PL[102]) interacting with the Leu152 containing hydrophobic cavity of VPS29. **B** The TBC1D13 interaction with Retromer involves three notable interfaces. View 1 shows the binding of a PL motif and extended beta-strand to the VPS29 subunit at the canonical site surrounding VPS29 Leu152. View 2 shows an extended beta-hairpin from TBC1D13 on the opposite face of VPS29 that also makes additional contacts with VPS35. Lastly, view 3 shows how VPS29 N-terminal Methionine Met1 (in standard isoform Q9UBQ0) is tightly buried in the complex with TBC1D13. **C, D** GFP-based co-immunoprecipitation (co-IP) of GFP-VPS29 wild-type (WT) and hydrophobic pocket mutants after transient transfection in HEK293T cells. Quantitation and statistical analysis of relative band intensity for the indicated proteins, normalised to GFP band intensity. $n = 3$ independent experiments. 1-way ANOVA with Dunnett's multiple comparison test, data presented as mean values relative to WT

and error bars represent standard deviation (SD). **E** ITC of TBC1D13 binding to Retromer in the absence or presence of competing cyclic peptide RT-D1. **F, G** GFP-based co-IP of GFP-TBC1D13 WT and P101A mutant after transient transfection in HEK293T cells. Quantitation and statistical analysis of relative band intensity for the indicated proteins, normalised to GFP band intensity. $n = 3$ independent experiments. Two-sided $t$ test analysis, data presented as mean values relative to WT and error bars represent SD. **H** Spinning-disk confocal images showing colocalization of transiently expressed GFP-TBC1D13 with endogenous VPS35 in Hela cells; wild-type (WT) and P101A mutant. Cells were fixed ~15 hrs after plasmid transfection, and images were acquired for cells with low to moderate GFP expression. **I** Quantitation and statistical analysis of colocalization between GFP-TBC1D13 and VPS35 after manual exclusion of the Golgi area and using ImageJ colocalization JACOP plugin, Manders coefficients M1/M2. $n = 3$ independent experiments. Two-sided $t$ test analysis, data presented as mean values relative to WT and error bars represent SD. Only changes with $p < 0.05$ are shown. Scale bar – 10μm.

**Table 2 | Thermodynamic parameters for the binding of Retromer and TBC1D13 by ITC**

| | $K_d$ (μM) | $\Delta H$ (kcal/mol) | $\Delta G$ (kcal/mol) | $-T\Delta S$ (kcal/mol) |
|---|---|---|---|---|
| hRetromer | | | | |
| hTBC1D13TBC | $6.3 \pm 1.2$ | $-8.1 \pm 1.4$ | $-7.1 \pm 0.1$ | $1.0 \pm 1.4$ |
| hTBC1D13TBC + RT-D1 | No binding detected | | | |

inner layer (*e.g.*, RAB7, SNX3, and various cargo proteins) and outer layer (*e.g.*, TBC1D5, ANKRD27, DNAJC13 and the WASH complex) of the Retromer coat complex and/or those proteins transiting through the Retromer retrieval sub-domain. Finally, for the corresponding core VPS35L α-solenoid of Retriever, such a clear separation between the amino-terminal and carboxy-terminal proximity proteomes is not observed. The recently described structure of Retriever and its association with the CCC complex to form the Commander super-assembly provides the molecular explanation for these data[20–22]. Here, the amino-terminus and carboxy-terminus of VPS35L reside in proximity to one another. Our proximity proteomic data therefore provides quantitative *in-cellulo* evidence consistent with the organisational features of Retriever and its assembly into Commander.

In addition to the validation of established structural and organisational features of Retromer and Retriever and their known accessory proteins, our data-rich proximity proteomic resource provides a wealth of molecular details of endosomal cargo retrieval and recycling. Our identification and preliminary validation of the augmin complex within the Retromer proximity proteome suggests a potential localised regulation of microtubule assembly during endosomal function or during a specialised role, such as in cell division[117]. Retromer has previously been linked to the γTuRC through its accessory protein ANKRD50[50], and endosomal dynamics and endosomal cargo retrieval and recycling are heavily reliant on plus-end and minus-end directed microtubule motor proteins[118–120]. Exploring a potential link of the endosomal Retromer retrieval sub-domain with the regulation of localised microtubule dynamics, especially within non-mitotic neurons, certainly warrants further investigation.

The Retromer proximity proteome has highlighted a close association with regulators of small GTPases, including well-established Retromer accessory proteins TBC1D5 and ANKRD27[77,82]. We have now extended these regulators to include DENND4A and DENND4C, TBC1D13, and a TBC1D1/TBC1D4 heterodimer, and in so doing, established mechanistic links between Retromer and the switching of RAB10 and RAB35; two RAB GTPases previously linked to the regulation of endosomal cargo recycling[85,88–90,92,96,101,121,122]. Our structural dissection of Retromer's association to these RAB10 and RAB35 regulators provides a mechanism to link sequence-dependent cargo selection and retrieval sub-domain organisation with the regulation of these

switches during endosomal cargo retrieval and recycling. In all cases, the accessory protein directly associates with Retromer through presentation of PL motifs to the same hydrophobic pocket in VPS29 that accommodates PL motifs from TBC1D5, ANKRD27 and the FAM21 subunit of the WASH complex[44,77,82,114]. Secondary associations with other features of VPS29 or VPS35 are required to stabilise the overall association to Retromer. With evidence that Retromer is required for the endosomal targeting of these RAB regulators, these data reveal a far greater complexity in the role of Retromer in regulating selected RAB GTPases than previously appreciated. In addition to its scaffolding roles in associating with cargo adaptors for the sequence-dependent recognition of cargo proteins and the WASH complex for aiding retrieval sub-domain organisation, Retromer should therefore also be viewed as a major endosomal hub for controlling the activation status and switching of specific RAB GTPases. This feature of Retromer appears to be specific in that, because of its structural architecture[20–22], Retriever is unable to associate with any of these RAB regulators.

Consistent with the direct association of Retromer to the RAB10 GEFs DENND4A and DENND4C and the RAB GAPs TBC1D1 and TBC1D4, functional analysis has identified a role for RAB10 in the endosomal retrieval and cell surface recycling of a classic Retromer cargo, the glucose transport GLUT1. It is important to note, however, that while this cargo transport phenotype is quantifiable and statistically significant, the level of phenotypic penetrance is far less than observed upon perturbation of SNX27 (the cargo adaptor for GLUT1) or Retromer[10]. This may reflect a level of redundancy in the routes that GLUT1 may take to recycle back to the cell surface once the essential SNX27 and Retromer-orchestrated retrieval fate decision has been made. Indeed, an element of this redundancy may stem from Retromer's hub-like ability to regulate several RABs linked to endosomal cargo recycling.

How are all these GEF and GAP interactions coordinated during Retromer's function to ensure successful cargo selection and transport? This is a very challenging question that ultimately will require more targeted analysis utilising the acquired structural information coupled with knock-in technology to precisely disrupt protein:protein interactions, and high-speed, 4D multi-spectral live imaging of endogenously tagged Retromer, cargo, and individual RAB regulators to reveal the timeline of RAB regulation across cargo entry and exit through the retrieval sub-domain. That said, it is important to note that based on expression levels from OpenCell (https://opencell.sf.czbiohub.org), HEK293T cells express far more Retromer than Retriever (VPS35 at 800 nM compared with VPS35L at 72 nM), and that VPS29A is expressed at sufficiently high enough excess (approximately 1100 nM) to be present in both complexes without necessarily having to compete between the two. Moreover, the Retromer-associated RAB regulators are expressed between 5-to-42-fold lower (DENND4A – 26 nM; VARP – 29 nM; TBC1D1 – 34 nM; DENND4C – 64 nM; TBC1D5 –

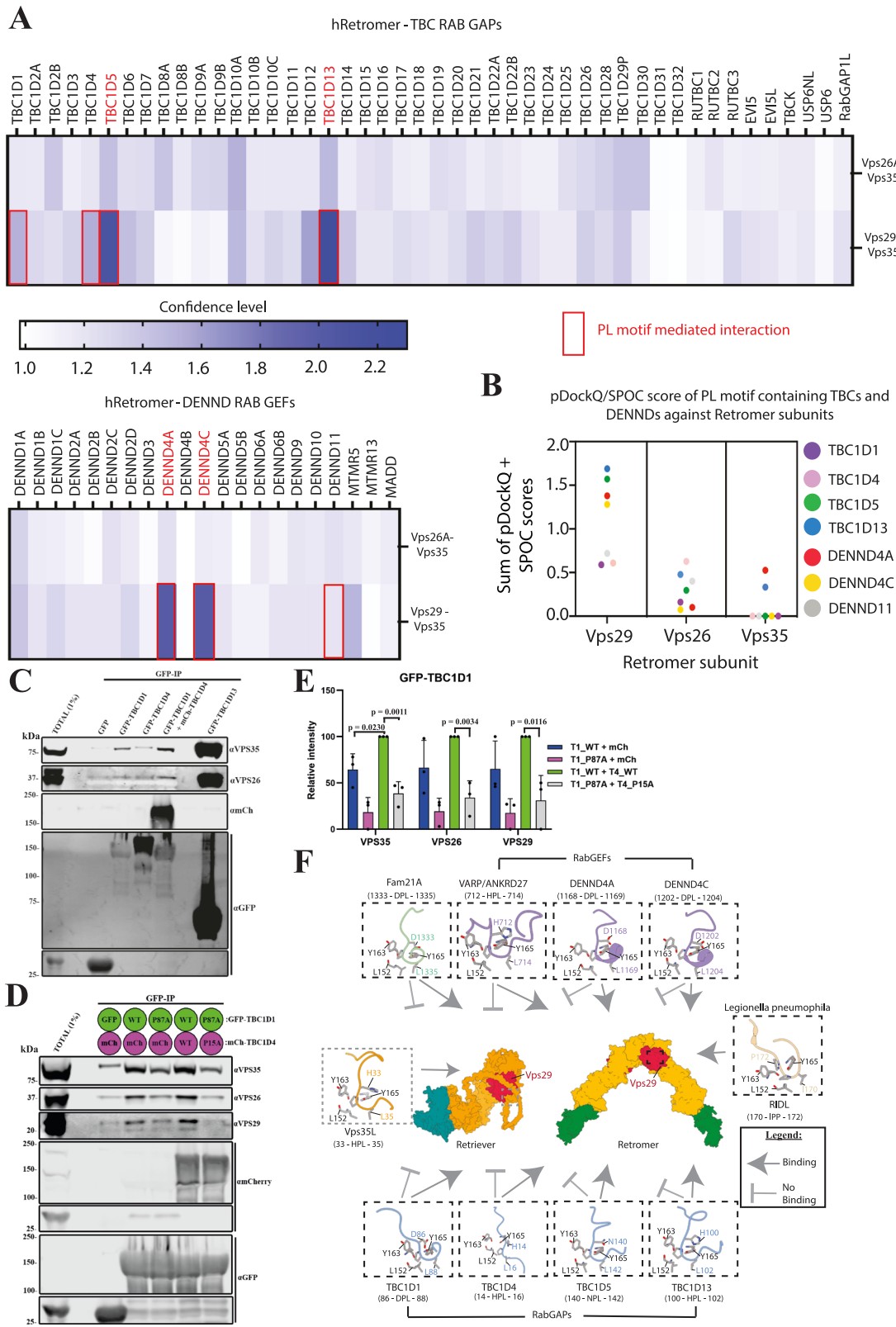

72 nM; TBC1D13 – 190 nM; TBC1D4 – 210 nM), establishing that the availability of VPS29 binding sites far exceeds the concentration of PL motif-containing proteins. The pseudo-helical array of Retromer arches therefore has the capacity to be heterogeneously labelled with RAB regulators and with the FAM21-containing WASH complex (FAM21A expressed at 180 nM).

The relative concentration of each individual regulator, their individual affinities derived from PL motif recognition and secondary recognition by VPS35, and additional low affinity binding to features of the local environment such as lipids, membrane geometry, F-actin and non-Retromer-interacting proteins, will provide the avidity that defines their enrichment and residency at the Retromer-demarcated

**Fig. 7 | Retromer as a hub for RAB GTPase switch regulation: VPS29 interaction screen and TBC1D1/4 interaction with Retromer. A**, **B** Retromer interactions with TBC GAP and DENND GEF proteins across the human proteome. VPS26A-VPS35 and VPS29-VPS35 assemblies or individual VPS29, VPS26A and VPS35 proteins were screened against all human proteins with TBC and DENN domains using Alpha-Fold2. The confidence of potential interactors was scored based on the sum of the interfacial PTM score (iPTM) averaged from three models, pDOCKQ and SPOC scores. In the TBC domain GAPs, TBC1D5 and TBC1D13 show high confidence. In the DENN domain GEFs, DENND4A and DENND4C were confidently predicted to bind to VPS29/Retromer. Others found to possess a PL motif but show low confidence in association with VPS29 binding are TBC1D1, TBC1D4 and DENND11. (**C**) Single experiment of GFP-based co-immunoprecipitation (co-IP) of GFP-TBC1D1, GFP-TBC1D4 or GFP-TBC1D13 after transient transfection in HEK293T cells. Co-

expression of GFP-TBC1D1 with mCherry-TBC1D4 increased pull-down of Retromer complex subunits. **D**, **E** GFP-based co-IP after transient transfection in HEK293T cells of wild-type (WT) and PL interacting-motive mutant versions of GFP-TBC1D1 (P87A) and mCherry-TBC1D4 (P15A). Quantitation and statistical analysis of relative band intensity for the indicated proteins, normalised to GFP band intensity. $n = 3$ independent experiments. Two-sided $t$ test analysis, data presented as mean values relative to WT and error bars represent SD. Only changes with $p < 0.05$ are shown. **F** Schematic summarising the interactions of Retromer with the identified TBC GAPs and DENND GEFs and the similarity with binding to the FAM21 subunit of the WASH complex and the RidL protein from *Legionella pneumophila*. The intra-molecular occlusion of the equivalent binding site in VPS29 when assembled in the Retriever complex prevents Retriever from binding to any of these proteins through these specific mechanisms.

retrieval sub-domain. Along with their relative catalytic activity, this will define the local activation state of RABs and the association or disassociation of their functional effectors. The dynamic instability within the Retromer accessory protein network is likely a feature in pathway progression and pathway fidelity that ensures directionality towards the successful biogenesis of a cargo-loaded transport carrier decorated with the appropriate RAB identity code, with post-translational modifications modifying and fine-tuning pathway progression by strengthening and weakening the avidity of Retromer network associations[123]. The most simplified model to describe a highly dynamic and highly complex set of associations, is that the co-residence of TBC1D5 and TBC1D13 to a pseudo-helical Retromer coated tubule would ensure that membrane exiting the retrieval sub-domain has RAB7 in an inactive state (*i.e.*, lacks late endosomal identity), and is devoid of active RAB35 to restrict association of its effector MICAL1 thereby preventing F-actin disassembly while simultaneously allowing ARF6 activation to aid recycling[97,124,125]. Association of DENND4A/C and VARP would drive activation of RAB10 and RAB21 and the acquisition of compartment identity and effector recruitment necessary for cargo delivery into onward recycling compartments[126]. As stated, this is certainly an over-simplification of a highly dynamic and complex set of regulatory steps and does not consider the potential distinct environment between SNX3-Retromer coated tubular profiles and RAB7-Retromer coated tubular profiles emerging from endosomes (*e.g.*, binding of RAB7-GTP to VPS35 may restrict access to the secondary binding required by some regulators thereby ensuring they preferentially associate with SNX3-Retromer arrays). Neither does it consider the possibility of preferential *trans* association of certain PL motif-containing accessory proteins such as VARP and FAM21 across pseudo-helical Retromer arches[44,82,114] over *cis* association with membrane-associated monomeric Retromer. A considerable amount of additional work will be required to build a detailed mechanistic understanding of the interface between Retromer and these RAB regulators during endosomal cargo recycling.

As for retrieval sub-domain dysfunction in human disease, it is worth noting that several RABs identified as being controlled by Retromer are substrates for LRKK2-mediated phosphorylation in response to disrupted lysosomal homoeostasis[127–129]. As the Parkinson's disease-associated VPS35(D620N) mutation leads to enhanced LRRK2 activation and target RAB phosphorylation[130], it is tempting to speculate that Retromer's role as a RAB regulatory hub may constitute a feedback controller in integrating lysosomal homoeostasis with endosomal retrieval sub-domain function. Exploring this potential point of connectivity may provide insight into Retromer and the retrieval sub-domain's neuroprotective role in Alzheimer's disease, Parkinson's disease and other neurodegenerative conditions.

Finally, it should be noted that while we have utilised CRISPR-Cas9 knockout and titrated lentiviral transduction of BioID-tagged rescue transgenes to allow as near to endogenous expression as technically possible, BioID tagging of the endogenous gene loci will allow

endogenous tagging of proteins to further validate and extend the present findings. The relatively poor kinetics of proximity labelling by BioID1 has limited our study to a steady-state analysis. While this has established the correct targeting strategy for each chosen protein and has revealed molecular insight, replacement of the BioID1 enzyme with more kinetically rapid proximity labelling reagents, such as APEX2, TurboID and AirID, will allow for time-resolve proximity proteomics. This will provide temporal access to the dynamics of the retrieval sub-domain make-up, organisation, and adaptation during stimulated cargo sorting. Finally, while our study has identified links between the retrieval sub-domain and SNARE proteins, the augmin complex, caveolae, and additional functionally interesting proteins, a great deal of further experimentation will be required to validate their localisation and functional role within the retrieval sub-domain. Equally, the molecular structures identified through AlphaFold analysis warrant future experimental studies to be fully verified.

## Methods
### Antibodies
Primary antibodies: β-Actin (Sigma-Aldrich, A1978; 1:5000 WB, 1:1000 IF), GAPDH (Sigma-Aldrich, G9545; 1:5000 WB), GAPDH (Sigma-Aldrich, G8795; 1:5000 WB), EEA1 (Cell Signalling; 3288S; 1:200 IF), GFP (Roche, 11814460001; clones 7.1/13.1; 1:1000 WB), mCherry/RFP (Abcam, 167453; 1:1000 WB), FLAG-M2 (Sigma, F1804; 1:1000 WB, 1:200 IF), GLUT1 (Abcam; ab115730; 1:1000 WB; 1:400 IF), LAMP1 (Developmental Studies Hybridoma Bank; AB_2296838; clone H4A3; 1:400 IF), VPS29 (Santa Cruz; D-1; sc-398874; 1:500 WB), VPS35 (Abcam, ab97545; 1:1000 WB, 1:400 IF), VPS35 (Antibodies.com, A83699; 1:400 IF), VPS26 (Abcam, ab23892; 1:1000 WB), VPS35L (Abcam, ab97889; 1:1000 WB), VPS26C (Sigma-Aldrich, ABN87; 1:1000 WB), VPS35L (Invitrogen, PA5-28553; 1:200 IF), SNX1 (BD Transduction Lab, 611482; 1:1000 WB, 1:200 IF), SNX1 (Proteintech, 10304-1-AP; 1:200 IF), SNX2 (BD Transduction Lab, 611308; 1:1000 WB), SNX5 (Abcam, ab180520; 1:1000 WB), SNX6 (Santa Cruz, sc-365965; 1:1000 WB), SNX17 (Proteintech, 10275-1-AP; 1:1000 WB), SNX17 (Sigma, HPA043867, 1:100 IF), SNX27 (Proteintech, 16329-1-AP; 1:1000 WB), SNX27 (Abcam, ab77799; 1:100 IF), HRS/HGS (Enzo Life Sciences, ALX-804-382-C050; 1:2000 WB, 1:100 IF), STAM (Proteintech, 12434-1-AP; 1:2000 WB, 1:100 IF), FAM21 (Gift from Dan Billadeau; 1:2000 WB), Integrin-α5 (Abcam, ab150361; 1:1000 WB, 1;200 IF), DENND4C (Sigma-Aldrich, HPA014917, 1:1000 WB, 1:500 IF), DENND4C (StressMarq Bioscience, SMC-610; 1:100 IF), DENND4A (Abcam, ab117758, 1:500 WB), HAUS2 (ThermoFisher, PA5-31258; 1:500 WB), HAUS6 (Proteintech, 16933-1-AP; 1:1000 WB), HAUS8 (Abcam, ab95970; 1:250 WB), CAV1 (Proteintech, 16447-1-AP; 1:1000 WB), CAVIN1 (Proteintech, 18892-1-AP; 1:1000 WB), TBC1D13 (ThermoFisher, PA5-61110; 1:2000 WB, 1:200 IF), ARHGAP1 (Proteintech, 11169-1-AP; 1:1000 WB, 1:100 IF), VAMP3 (Proteintech, 10702-1-AP; 1:1000 WB, 1:200 IF), NF1 (Proteintech, 27249-1-AP; 1:1000 WB), ARFGEF1 (Abcam, A44423,1:1000 WB), TBC1D4 (Proteintech, 68063), KIDINS220 (Proteintech, 21856-1-AP; 1:1000 WB), ATP7A (Santa Cruz, sc-376467, 1:1000 WB), CTR1

(Abcam, ab129067, 1:1000 WB), LAT1 (Cell Signalling, 5347 s, 1:1000 WB), TBC1D5 (Abcam, 203896, 1:1000 WB), VARP/ANKRD27 (Proteintech, 24034-1-AP, 1:1000 WB), RAB10 (Abcam, ab237703, 1:1000 WB, 1:200 IF). Secondary antibodies for WB: 680 nm and 800 nm donkey anti-mouse and anti-rabbit fluorescent secondary antibodies (Invitrogen, A-21057, A3275; 1:20,000). Secondary antibodies for IF: 405, 488, 568 and 647 nm AlexaFluor-labelled anti-mouse, anti-rabbit or anti-goat (Invitrogen; 1:400).

## Plasmids and molecular cloning

cDNA codifying for the proteins expressed were cloned into pEGFP/mCherry-C1/N1 transient vectors or pLVX lentiviral vectors by restriction enzyme digestion and ligation or Gibson Assembly reaction. Sequence for BioID1, preceded by a FLAG tag, was fused by a flexible linker to the N or C terminus of selected proteins and incorporated in the pLVX-CMV-MCS-PGK-Puro vector. CRISPR-Cas9 gRNAs were designed as a couple of opposing DNA primers which were phosphorylated, annealed and cloned into pSpCas9(BB)−2A-Puro (pX459) BbsI digested. Resulting plasmids were transformed in XL1-Blue (Agilent) or NEB 5-alpha HE (NEB) *E. coli* strains and grown on selecting media, purified and validated by Sanger sequencing. gRNAs sequences are shown in Supplementary Table S1 and list of plasmids in Supplementary Table S2.

For bacterial expression, full-length human VPS35 and VPS26A with an N-terminal Hi-tag were cloned into the pET28a vector as described previously[106,131]. Human VPS29 was cloned into the pGEX4T-2 vector with a linker between the start of the gene and the thrombin recognition site to facilitate the tag cleavage. DNA encoding the bacteria expression optimised human full-length TBC1D13 was synthesised and cloned into the pGEX6P-1 vector by Gene Universal. All DNA constructs were verified using DNA sequencing.

## Cell culture, transfection and lentiviral production

HeLa and HEK293T cell lines were sourced from ATCC. Authentication was from the ATCC, and additionally, the HeLa line was authenticated by Eurofins services. Cells were grown in DMEM (Sigma-Aldrich) supplemented with 10% (v/v) FCS (Sigma-Aldrich) and penicillin/streptomycin (Gibco, USA) and grown in humidified incubators at 37 °C, 5 % $CO_2$. FuGENE HD (Promega, USA) or Lipofectamine LTX (Invitrogen) were used for transient transfection of DNA according to the manufacturer's instructions. Alternatively, for immunoprecipitation or viral production, cells were transfected with polyethylenimine (PEI) (Sigma-Aldrich) in 3:1 PEI:DNA weight ratio.

For the generation of CRISPR-Cas9 KO HeLa cells, cells were transfected with pX459 plasmid coding for the gRNA against the gene of interest, as well as the puromycin resistance. One day after transfection, cells were subjected to puromycin selection for 24 h. After 3-4 days recovering without puromycin, the pooled population of cells was subjected to lysis and WB to confirm the KO. Once the gRNA was validated, the experiment was repeated, but after puromycin selection, cells were diluted for clonal selection in 96-well plates with IMDM (Sigma-Aldrich). After 3-4 weeks, clonal populations were escalated and checked by WB. VPS35KO, SNX27KO, SNX17KO, and VPS35LKO clonal cell lines used in this study were previously characterised[19,20,31].

For transient GFP/mCherry-based immunoprecipitations, HEK293T cells in 15 cm plates at 70% confluency were transfected with GFP/mCherry plasmids using the PEI procedure, and after 24–48 h, cultures were lysed for protein purification. For lentiviral production, HEK293T cells in 15 cm plates at 70% confluency were co-transfected with the pLVX and the helper plasmids PAX2 and pMDG2 using PEI. After 4 h incubation, the media was replaced, and 48 h after transfection, the media containing the virus was harvested, spun down and filtered through 0.45 μm filters before being aliquoted and stored at − 80 °C.

## Immunoprecipitation and quantitative western blot analysis

For every kind of protein extraction, buffers were pre-chilled, and cells and samples placed on ice. For basic western blot (WB), cells were lysed in ~ pH 7.2 phosphate buffered saline (PBS) with 1% (v/v) Triton X-100 and protease inhibitors (Thermo Fisher, A32955). The protein concentration was determined with a bicinchoninic acid (BCA) assay kit (Thermo Fisher, 23225) and equal amounts were denaturised (LDS Sample buffer (Thermo Fisher, NP0008), 3% β-mercaptoethanol (Sigma-Aldrich, M6250), 10 min at 95 °C) and resolved by SDS-PAGE on NuPAGE 4–12% precast gels (Invitrogen, NP0336BOX). Protein samples were then transferred onto methanol-activated polyvinylidene fluoride (PVDF) membrane (Immobilon-FL membrane, pore size 0.45μm; Millipore, IPFL00010). The membrane was blocked, then sequentially labelled with primary and secondary antibodies. Fluorescence detected by scanning with a LI-COR Odyssey scanner and Image Studio analysis software (LI-COR Biosciences). Typically, we performed WB in where a single blot is simultaneously probed with antibodies against 2 proteins of interest (distinct antibody species), followed by visualisation with the corresponding secondary antibodies conjugated to distinct spectral dyes.

For GFP/mCherry-based co-immunoprecipitations (co-IP), cells were lysed in IP buffer (50 mM Tris-HCl, 0.5% (v/v) NP-40, and protease inhibitors) and centrifuged 10 min at 16.000 x *g*. An aliquot of the cleared lysate was retained to represent the whole cell fraction, and the rest added to pre-washed (in IP buffer) GFP/mCherry-trap beads (ChromoTek, gta-20/rta-20) for rocking 1 h at 4 °C. After, beads were washed 3 times with IP buffer by rounds of re-suspension and pelleting. Finally, the buffer was removed, and the beads were then either stored at − 20 °C or processed for SDS-PAGE analysis. For proteomic analysis, detergent in final washes was sequentially removed, and samples processed immediately.

## BioID1 protein-proximity labelling

Cells stably expressing the protein of interest tagged with BioID1 were grown in 15 cm plates. When confluency reached 70%, the media was changed by DMEM supplemented with 50 μM biotin (Sigma-Aldrich) and incubated typically for 24 h. Cells were washed twice in PBS and lysed in RIPA buffer (50 mM Tris-HCl pH 7.5, 150 mM NaCl, 0.1% SDS (v/v), 0.5% (w/v) Sodium Deoxycholate, 1% (v/v) Triton X-100 and protease inhibitors (Roche, 05892970001)). Lysates were centrifuged 10 min at 16.000 x *g* and the supernatants were collected. The protein concentration was determined with a BCA assay kit. An aliquot of the cleared lysate was retained to represent the whole cell fraction, and the rest added to pre-washed (in RIPA buffer) streptavidin sepharose beads (Sigma-Aldrich, Cytiva 17511301) for rocking 2 h at 4 °C. Later, beads were sequentially washed by re-suspension and pelleting: twice in RIPA buffer, once in 1 M KCl, once in 100 mM $Na_2CO_3$, once in 2 M Urea in 10 mM Tris-HCl pH 8 and twice in RIPA buffer. Finally, whole cell fractions and beads were either stored at − 20 °C or denaturised (LDS Sample buffer, 3% β-mercaptoethanol, 20 mM DTT (Thermo Fisher, 20290), 2 mM biotin, 10 min at 95 °C) and resolved by SDS-PAGE. For proteomic analysis, beads were washed once more with RIPA buffer without detergents, and samples processed immediately.

## Surface biotinylation

All steps were carried out on ice to prevent surface protein internalisation. Just prior to starting the biotinylation labelling, membrane impermeable Sulfo-NHS-SS-Biotin (Thermo Fisher, 21331) was dissolved at a final concentration of 0.2 mg/ml in PBS adjusted to pH 7.8. Cells were washed twice in PBS before being incubated with Sulfo-NHS-SS-Biotin for 20 min. Later, cells were washed twice in quenching buffer (50 mM Tris-HCl pH 7.5, 100 mM NaCl) and incubated on it for 10 min. Then, cells were lysed in PBS, 1% Triton X-100 plus protease inhibitor cocktail (Thermo Fisher, A32955). Lysates were centrifuged

10 min at 16.000xg and the supernatants were collected. The protein concentration was determined by BCA. An aliquot of the cleared lysate was retained to represent the whole cell fraction and the rest added to pre-washed (in PBS plus 1% Triton) streptavidin sepharose beads for rocking 1 hr at 4 °C. Afterwards, beads were sequentially washed by resuspension and pelleting: three times in PBS plus 1% Triton plus 1.2 M NaCl. Finally, whole cell fractions and beads were either stored at − 20 °C or denaturised (LDS Sample buffer, 3% β-mercaptoethanol, 20 mM DTT, 2 mM biotin, 10 min at 95 °C) and resolved by SDS-PAGE.

### Immunofluorescence Microscopy and Analysis
HeLa cells were seeded onto 13 mm coverslips. Cells were fixed in 4% (v/v) paraformaldehyde (PFA) (Pierce, 28906) in PBS for 20 min and permeabilised in 0.1% (v/v) Triton X-100 in PBS (Sigma-Aldrich), or in 0.1% (w/v) saponin (sigma-Aldrich) in PBS when labelling LAMP1 compartments, for 5 min followed by blocking with 2% (w/v) BSA (Sigma, 05482) in PBS for 30 min. Coverslips were stained with primary antibodies for 1 hr followed by secondary antibodies for 30 min, then mounted onto glass microscope slides with Fluoromount-G (Invitrogen, 00-4958-02).

Confocal microscope images were taken on a Leica SP5-II confocal laser scanning microscope attached to a Leica DMI 6000 inverted epifluorescence microscope or a Leica SP8 confocal laser scanning microscope attached to a Leica DM l8 inverted epifluorescence microscope (Leica Microsystems), with a 63x UV oil immersion lens, numerical aperture 1.4 (Leica Microsystems, 506192). For the Leica SP8 microscope, 'lightning' adaptive image restoration was used to generate deconvolved representative images. Colocalization analysis was performed using Volocity 6.3 software (PerkinElmer) or JACOP plugin from ImageJ-FIJI software. Spinning disk high-resolution images were acquired on an IXplore SpinSR system (Olympus) consisting of an IX83 microscope frame with a $CO_2$ and temperature chamber (Olympus), SoRa W1 twin cam spinning disk unit (Yokogawa) and two back-thinned Fusion BT sCMOS cameras (Hamamatsu). A 488 nm or 561 nm laser provided excitation for the green and red channels, respectively. The SoRa disk with an additional 3.2x magnification changer was used for imaging with excitation light focused to the sample using a 60x/ 1.4NA oil immersion lens. The system operated in simultaneous twin-cam mode, and the desired fluorescence was selected through the use of either a 525/50 nm or 617/73 nm bandpass filter for the green or red channel, respectively.

### TMT Labelling and High pH reversed-phase chromatography
Proteomic experiments were performed with isobaric tandem mass tagging (TMT) coupled to nanoscale liquid chromatography joined to quantitative tandem mass spectrometry (nano-LC-MS/MS). Bead samples were immediately reduced with 10 mM TCEP (55 °C for 1 h), alkylated with 18.75 mM iodoacetamide (room temperature for 30 min) and then digested from the beads with trypsin (2.5 µg trypsin; 37 °C, overnight). After the digestion, the resulting peptides were labelled with TMT 10 or 8 plex reagents according to the manufacturer's protocol (Thermo Fisher). The samples were pooled and evaporated to dryness, resuspended in 5% formic acid and next desalted using a SepPak cartridge according to the manufacturer's instructions (Waters™). Eluate from the SepPak cartridge was again evaporated to dryness and resuspended in buffer A (20 mM $NH_4OH$, pH 10) before to be fractionated by high pH reversed-phase chromatography using an Ultimate 3000 liquid chromatography system (Thermo Fisher). Shortly, the sample was loaded onto an XBridge BEH C18 Column (130 Å, 3.5 µm, 2.1 mm × 150 mm, Waters™) in buffer A and the peptides were eluted with an increasing gradient of buffer B (20 mM Ammonium Hydroxide in acetonitrile, pH 10) from 0–95% over 60 min. 5 fractions were generated, evaporated to dryness, and finally resuspended in 1% formic acid to get them ready for analysis by nano-LC MSMS using an Orbitrap Fusion Tribrid mass spectrometer (Thermo Fisher).

### Nano-LC mass spectrometry
The resulting fractions were further fractionated with an Ultimate 3000 nano-LC system in line with an Orbitrap Fusion Tribrid mass spectrometer (Thermo Scientific). Concisely, peptides in 1% (vol/vol) formic acid were injected onto an Acclaim PepMap C18 nano-trap column (Thermo Scientific). After washing with 0.5% (v/v) acetonitrile and 0.1% (v/v) formic acid, peptides were resolved on a 250 mm × 75 µm Acclaim PepMap C18 reverse phase analytical column (Thermo Scientific) over a 150 min organic gradient, including 7 gradient segments (1–6% solvent B over 1 min, 6–15% B over 58 min, 15–32% B over 58 min, 32–40% B over 5 min, 40–90%B over 1 min, held at 90% B for 6 min and then reduced to 1% B over 1 min) at a flow rate of 300 n/min. Solvent A was 0.1% formic acid, and Solvent B was aqueous 80% acetonitrile in 0.1% formic acid (v/v). Peptides were ionised by nano-electrospray ionisation at 2.0 kV using a stainless-steel emitter with an internal diameter of 30 µm (Thermo Scientific) and a capillary temperature of 275 °C. All spectra were acquired using an Orbitrap Fusion Tribrid mass spectrometer controlled by Xcalibur 3.0 software (Thermo Scientific) and operated in data-dependent acquisition mode using an SPS-MS3 workflow. FTMS1 spectra were collected at a resolution of 120,000, with an automatic gain control (AGC) target of 200,000 and a maximum injection time of 50 ms. Precursors were filtered with an intensity threshold of 5000, according to charge state (to include charge states 2–7) and with monoisotopic peak determination set to peptide. Formerly interrogated precursors were omitted using a dynamic window (60 s +/− 10 ppm). The MS2 precursors were isolated with a quadrupole isolation window of 1.2 m/z. ITMS2 spectra were collected with an AGC target of 10,000, CID collision energy of 35% and max injection time of 70 ms. For FTMS3 analysis, the Orbitrap was operated at 50,000 resolution with an AGC target of 50,000 and a maximum injection time of 105 ms. Precursors were fragmented by high-energy collision dissociation (HCD) at a normalised collision energy of 60% to ensure maximal TMT reporter ion yield. Synchronous Precursor Selection (SPS) was enabled to include up to 10 MS2 fragment ions in the FTMS3 scan.

### Statistics and bioinformatic analysis
Proteomic raw data files were processed and quantified using Proteome Discoverer software v2.1 (Thermo Scientific) with the SEQUEST HT algorithm and searching against the UniProt Human database (downloaded 2021-01-14; 178486 sequences). Peptide precursor mass tolerance was set at 10 ppm and MS/MS tolerance at 0.6 Da. Search criteria included oxidation of methionine (+ 15.995 Da), acetylation of the protein N-terminus (+ 42.011 Da) and Methionine loss plus acetylation of the protein N-terminus (− 89.03 Da) as variable modifications and carbamidomethylation of cysteine (+ 57.021 Da) and the addition of the TMT mass tag (+ 229.163 Da) to peptide N-termini and lysine as fixed modifications. Searches were performed with full tryptic digestion, and a maximum of 2 missed cleavages were allowed. The reverse database search option was enabled, and all data was filtered to satisfy a false discovery rate (FDR) of 5%. Where proteins were identified and quantified by an identical group of peptides as the master protein of their protein group, these are designated 'candidate master proteins'. Next, we used the annotation metrics for candidate master proteins retrieved from Uniprot to select the best annotated protein, which was then designated as the master protein. This enables us to infer biological trends more effectively in the dataset without any loss in the quality of identification or quantification.

For statistical analysis of differential protein abundance between conditions, standard two-sided t-test analysis were used. Volcano plots were generated with VolcanoseR2 webapp[132] or GraphPad Prism 9 software (LaJolla, CA). Gene ontology analysis of the proteins identified was performed using Metascape 3.5 webapp[54] and Cytoscape 3.9 software to represent pathway enrichment and DisGeneNET category enrichment, using a hypergeometric test and Benjamini-

**Table 3 | Summary of crystallographic structure determination statistics**

| Data collection statistics | mVps29 – hDENND4C$_{1189-1206}$ |
|---|---|
| PDB ID | 8VOD |
| Beamline | MX2 |
| Space group | P 3$_2$ 2 1 |
| Resolution (Å) | 34.13 – 2.35 |
|  | (2.43 – 2.35) |
| a, b, c (Å) | 42.90, 42.90, 172.74 |
| α, β, γ (°) | 90.00, 90.00, 120.00 |
| Data collection temp. (K) | 100 |
| Wavelength (Å) | 0.95365 |
| Total observations | 16,359 (1561) |
| Unique reflections | 8279 (785) |
| Completeness (%) | 99.7 (100.0) |
| R$_{merge}$$^+$ | 0.028 (0.195) |
| R$_{pim}$* | 0.028 (0.195) |
| CC1/2 | 0.998 (0.891) |
| <I/σ(I)> | 9.87 (2.35) |
| Multiplicity | 2.0 (2.0) |
| Molecule/asym | 1 |
| Refinement statistics | – |
| R$_{work}$/R$_{free}$ (%)$^{\#}$ | 18.78/25.22 |
| No. protein atoms | 1590 |
| Waters | 26 |
| Wilson B (Å$^2$) | 49.33 |
| Average B (Å$^2$)^ | 53.92 |
| Protein | 53.84 |
| Water | 48.12 |
| rmsd bonds (Å) | 0.008 |
| rmsd angles (°) | 1.07 |
| Ramachandran plot: | – |
| Favoured/outliers (%) | 95.38/0.51 |

Values in parentheses refer to the highest resolution shell. $^+$R$_{merge}$ = Σ|I – <I>|/Σ<I>, where I is the intensity of each individual reflection. *R$_{pim}$ indicates all I$^+$ & I$^-$. $^{\P}$R$_{work}$ = Σh|F$_o$ - F$_c$| / Σ$_l$ |F$_o$|, where F$_o$ and F$_c$ are the observed and calculated structure-factor amplitudes for each reflection h. $^{\#}$R$_{free}$ was calculated with 10% of the diffraction data selected randomly and excluded from refinement. ^Calculated using Baverage.

Hochberg correction. Dot plot diagrams were generated using the platform prohits-viz.org[55] to better visualise the quantitative enrichment and statistical significance of each identified protein across all data sets. The mass spectrometry proteomics data have been deposited to the ProteomeXchange Consortium via the PRIDE partner repository with the dataset identifier PXD061918.

**Peptide synthesis**
The human DENND4C peptide (1189 – AKVVQREDVETGLDPLSL – 1206) and the L1204A mutant (1189 – AKVVQREDVETGLDPASL – 1206) were made by solid phase peptide synthesis in house, purified by reverse phase HPLC and purity assessed by mass spectrometry. VPS29 binding cyclic peptide RT-D1 (Ac-yIIDTPLGVFLSSLKRC-NH2) was synthesised and prepared as described previously[106].

**Recombinant protein expression and purification**
All the bacteria constructs were expressed in BL21 (DE3) cells using the autoinduction method. In brief, cultures grown in complex medium were incubated at 37 °C for 4 h, followed by overnight growth at 20 °C before harvesting for purification. To obtain the Retromer trimer, the GST-tagged Vps29 and Vps35 co-expressed cell pellet was mixed with His-Vps26A and resuspended in lysis buffer containing 50 mM Tris-HCl

pH 7.5, 200 mM NaCl, 2 mM 2-Mercaptoethanol, 50 μg/ml benzamidine and DNase I before being processed through a Constant System TS-Series cell disruptor. The cell pellet containing either GST-tagged VPS29 or GST-tagged hTBC1D13 was resuspended and lysed following the same procedure as for Retromer. In all cases, the soluble fractions clarified by centrifugation were loaded onto pre-equilibrated glutathione sepharose (GE healthcare) for initial purification. For Retromer, an additional Talon® resin purification step was performed to ensure the correct stoichiometry. GST-tag removal was carried out co-column overnight by adding thrombin for Human Retromer and Pre-Scission protease for TBC1D13. The GST-tag removed fraction was further purified by size-exclusion chromatography (SEC) using a Superdex 200 (16/600) column (GE Healthcare) equilibrated with buffer containing 50 mM Tris-HCl pH 7.5, 200 mM NaCl, 2 mM β-ME.

**Isothermal titration calorimetry (ITC)**
The binding affinities between Retromer, DENND4C and TBC1D13 were determined using a Microcal PEAQ-ITC (Malvern) at 25 °C. For Retromer and DENND4C, 1 mM of either the native or L1204A mutant DENND4C$_{1189-1206}$ peptide was titrated into 14 μM Retromer. The effect of cyclic peptide was assessed by pre-mixing 140 μM RT-D1 into 14 μM Retromer before titrating with 1 mM of native DENND4C$_{1189-1206}$ peptide. For the binding experiment between Retromer and TBC1D13, 530 μM of TBC1D13 was titrated into 14 μM Retromer. The effect of the cyclic peptide was tested using the same concentration of RT-D1 as described above. All ITC experiments consisted of an initial 0.4 μl (not used in data processing), followed by 12 serial injections of 3.22 μl each with 180 sec intervals. The resulting titration data were integrated with the Malvern software package by fitting and normalised data to a single-site binding model. This yielded the thermodynamic parameters for all binding experiments, including the dissociation constant (Kd), enthalpy (ΔH), Gibbs free energy (ΔG) and -TΔS. The stoichiometry (N) was refined initially, and if the value was close to 1, then it was set to exactly 1.0 for calculation. The data presented are the mean of triplicate titrations for each experiment.

**Crystallisation and structure determination**
Crystals of VPS29 – DENND4C$_{1189-1206}$ peptide complex were grown by hanging drop vapour diffusion method at 20 °C in a condition containing 0.1 M Tris-HCl pH 8.5, 0.2 M MgCl$_2$, 30% PEG4000. Best quality crystal was obtained by adding a 3-fold molar excess of DENND4C$_{1189-1206}$ peptide into the purified mVps29, reaching a final protein concentration of 12 mg/ml, with a protein-to-reservoir drop ratio of 2:3. Crystals grew after 10 days were cryoprotected in reservoir solution supplemented with 10% glycerol. X-ray diffraction data were collected at 100 K at the Australian Synchrotron MX2 beamline. The collected data was indexed and integrated using AutoXDS[133] and scaled using Aimless (version 0.7.4)[134]. The phase was solved by molecular replacement using Phaser (version 2.5.6) (Mccoy et al., 2007) with the native mVps29 structure as the initial mode template. After refinement, the electron density corresponding to the DENND4C peptide was clearly visible. The refinement was performed using Phenix (version 1.19.2)[135], with inspection of the resulting model in Coot (version 0.8.9.2), guided by the 2F$_o$ – F$_c$ map and F$_o$ – F$_c$ difference map. Molprobity[136] was used to assess the geometry quality of the refined model, and molecular figures were generated using PyMOL (version 2.5.5). Data collection and refinement statistics are summarised in Table 3. The VPS29 – DENND4C$_{1189-1206}$ structure has been deposited at PDB with identification code 8VOD.

**Protein structural prediction, modelling and visualisation**
All protein models were generated using AlphaFold2 Multimer implemented in the Colabfold interface available on the Google Colab platform. Due to the protein length, Alphafold3 was used to screen for potential interactions between full-length Retromer and TBCs/

DENNDs. In the case of the Alphafold2 modelling experiment, Colab-Fold was executed using default settings, where multiple sequence alignments were generated with MMseqs2. For all final models displayed in this manuscript, structural relaxation of peptide geometry was performed using AMBER. For all modelling experiments, we assessed (i) the prediction confidence measures (pLDDT and inter-facial iPTM scores), (ii) the plots of the predicted alignment errors (PAE) and (iii) backbone alignments of the final structures. The confidence level of Retromer interactions with TBCs/ DENNDs were calculated using the sum of IPTM score generated by Alphafold3 (full-length Retromer) and Alphafold2 (sub-complex and individual subunits). For clarity, the confidence level score was differentiated into two groups depending on the Retromer subcomplex (VPS26A – VPS35 and VPS29 – VPS35). SPOC score of Retromer individual subunits – TBCs/DENNDs Alphafold2 models was calculated using the web-based tool (https://predictomes.org/tools/spoc/) with default settings described previously[137]. Similarly, pDOCKQ score was obtained using a previously established method[103]. All structural images were made with Pymol (Schrodinger, USA; https://pymol.org/2/) and ChimeraX (version 1.6.1) (see Supplementary Table S3 for ModelArchive deposited models).

## Statistics & reproducibility

Statistics from western blots and confocal microscopy from a minimum of 3 independent experimental repeats was generated and represented using GraphPad Prism 9 software (LaJolla, CA). Graphs were plotted representing the mean value ± the standard deviation (SD) for each experimental condition. n represents the number of independent experimental repeats. Experiment-specific statistical details are included in the corresponding figure caption or in the Source Data file.

## Reporting summary

Further information on research design is available in the Nature Portfolio Reporting Summary linked to this article.

## Data availability

Data generated in this study is shown in the published article, including Supplementary Figs. 1–12. The mass spectrometry proteomics data have been deposited to the ProteomeXchange Consortium via the PRIDE partner repository with the dataset identifier PXD061918. Thermodynamic parameters from ITC experiments are included in Table 1 and 2. The crystallographic structure determination statistics are summarised at Table 3. CRISPR Cas 9 gRNAs are listed in Supplementary Table S1. Reagents and resources are listed in Supplementary Table S2. AlphaFold models have been deposited at ModelArchive with the ID numbers included in Supplementary Table S3. Unprocessed western blots and statistical data is included in this paper as the Source Data file. Other ascension codes used in this paper: 7BLN [https://doi.org/10.2210/pdb7BLN/pdb], 8VOD [https://doi.org/10.2210/pdb8VOD/pdb]. Source data are provided in this paper.

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

## Acknowledgements

We thank the Wolfson Bioimaging Facility at the University of Bristol for their support. Work in the Cullen laboratory is supported by the Wellcome Trust (104568/Z/14/Z and 220260/Z/20/Z), the Medical Research Council (MR/L007363/1 and MR/P018807/1), the Lister Institute of Preventive Medicine, and the award of a Royal Society Noreen Murray Research Professorship to P.J.C. (RSRP/R1/211004). We acknowledge the use of the University of Queensland Remote Operation Crystallisation and X-ray (UQ ROCX) Facility and the assistance of K.A. Arachchige. X-ray data were collected on the MX2 beamline at the Australian Synchrotron. B.M.C. is supported by an Investigator Grant from the National Health and Medical Research Council (APP2016410). C.A.P. was initially supported by Beca Fundación Ramón Areces Estudios Postdoctorales en el Extranjero.

## Author contributions

BioID and cell biology analysis: C.A-P. and A.J.E. Biophysics, structural analysis and AlphaFold2: K.E.C., Q.C. and M.L. Proteomics and Bioinformatic analysis: P.A.L. and K.J.H. Manuscript Writing – 1st draft: C.A-P., K.E.C., B.M.C. and P.J.C; Final Version: all authors. Initial Concept: C.A.-P. and P.J.C. Concept Development: all authors. Funding and Supervision: K.A.W., B.M.C. and P.J.C.

## Competing interests

The authors declare no competing interests.
