## [Transparent Peer Review file · Nature Communications]

Mapping of endosomal proximity proteomes reveals Retromer as a hub for RAB GTPase regulation

Corresponding Author: Professor Peter Cullen

Version 0:

Reviewer comments:

Reviewer #1

(Remarks to the Author)

The manuscript by Antón-Plágaro et al. describes the endosomal proximity-dependent proteomics of Retromer and Retriever, their cargo adaptors, and an ESCRT-degradative sub-domain protein. By combining proximity-labeling proteomics, X-ray crystallography, and AlphaFold2, the authors provided a proteomic landscape of endosomal sorting sub-domains and identified the previously unappreciated interactions between Retromer and RAB10&RAB35 regulators.

This is a well-written, well-designed, and controlled study. All experiments were conducted at a high level and the presented results generally support the conclusions. The study describes a workflow for proteomics analysis of endosomal retrieval and recycling of integral cargo proteins and provides novel insight into the role of Retromer in neuroprotection. In my view, the content and quality of this study make it suitable for publication in Nature Communications.

Line 118-: I appreciate that the authors carefully generated stable cell lines (by knocking our endogenous genes and by selecting clones expressing as near to endogenous levels of the chimera proteins) and extensively validated the localization and function of each chimera. Also, the two negative controls (cytosolic BioID and TOM70-RFP-BioID) are a good addition. I also like that the authors mentioned/discussed known interacting proteins with baits that could not be captured in this study. Overall, preparation/optimization for BioID cell lines and following proteomic analysis were conducted at a high level.

Figure 1-3: It would be hard for readers to follow the relationships between bait (BioID) proteins and their interactions. A cartoon(s) (like Figure 1B) or interaction networks must be very helpful to better understand an overview of the localization and topology of bait proteins and proximity-dependent interaction partners. Line 159: Is "50 mM" biotin correct?

Line 194: Mention how to correct batch effect for comparing different TMT measurements (if you performed).

Line 231: Could this be due to the relatively high-intensity signal of SNX27 in the negative control (cyto control), which made SNX27 less enriched in the experimental condition? A choice of negative control is sometimes a key to identifying bona fide interactions.

Along the same line, BioID is a relatively slow reaction (~12 h) compared to APEX2/HRP (~min), TurboID (~10 min), AirID (~30 min). Please comment about pros&cons including the possibility of loss of transient interactions with BioID.

Line 891: Is FDR 5% at PSM, peptide, or protein level? Explain why you adopted the 5% FDR (at PSM level?), though most of the studies use 1% FDR.

Table: In the excel file, some functions remained (e.g., $-\text{LOG}_{10}([\text{Stats_X27}/\text{Cyto_T-Test}])$), keep only values.

Deposit the proteomic raw files to a repository (e.g., PRIDE).

Reviewer #2

(Remarks to the Author)

The article presents a comprehensive map of the endosomal proximity proteome, offering insights into the molecular networks governing endosomal trafficking. By employing carefully designed BioID techniques, this study identifies the

association of Retromer with various previously uncharacterized components. Notably, its interaction with Rab GTPase regulators, including DENND4 and TBC1Ds, was extensively validated through biochemical and cellular analyses, as well as crystallography and AlphaFold predictions.

This work is an excellent example of how AlphaFold models can be utilized to study protein-protein interactions. The proteomic data on the endosomal sorting system in this study could be a valuable resource and guide for future research on Retromer components. The manuscript is well-written and contains critical analyses and discussions.

I have no major criticisms of this manuscript but offer some minor suggestions:

A significant portion of the protein-protein interaction analysis is based on AlphaFold predictions. Specifically, Figures 4 and 6 contain key interpretations of these predicted models. I suggest that the authors provide the AlphaFold models used for these figures as PDB or PyMOL files in the supplementary data, allowing readers to examine the models more closely.

The determination of the crystal structure and model building was well executed. However, in Table 3 on page 78, the Rmerge value for the highest resolution shell is very high, and the $I/\sigma(I)$ ratio is only 0.99. This suggests that the highest resolution shell contains no useful data, and the resolution limit has been overestimated. Additionally, the deposited electron density (ED) maps in the PDB (ID: 8VOD) do not show the typical quality expected for a 2.12 Å resolution dataset. To ensure accuracy, the data should be trimmed to a lower resolution where $I/\sigma(I)$ is greater than 2 or 3.

Reviewer #3

(Remarks to the Author)

Comments on experimental structure description for Antón-Plágaro et al.:

while otherwise technically results appear to be correct, the methods paragraph for crystallography should be completely rewritten, the language needs to be checked (English) and some descriptions need a clarification: peptide complex crystallization what is meant by "3 molar excesses"? 3 molar = 3 M? Perhaps the writer should rather refer to the molar ratio of the two molecules.

None of the text (grammar) below makes sense: "Best quality crystal was obtained by adding 3 molar excesses of DENND4C1189-1206 peptide into the purified mVps29 to a final concentration of 12 mg/ml with a protein-reservoir drop ratio of 2:3. Crystals grew after 10 days were cryoprotected in reservoir solution with additional 10% glycerol. X-ray diffraction data were carried out at the Australian Synchrotron MX2956 beamline at 100 K. "

This appears to be just incomplete sentences badly written.

This paragraph should be checked by native / fluent speaker before it is possible to accept. (e.g data is not carried out anywhere, the data is collected). "crystal" should be "crystals" most likely. AutoXDS (ie XDS) is missing a reference.

Difference map and refinement description is wrong, or something strange was done. please correct the whole paragraph to exact and sensible description. You cannot use Fo-Fc maps (alone) to build and inspect a model.

Table 3 should include the wavelength (Å) used measurements, as well as the beam line used.

PDB ID is missing please add these, as a side note refinement should have included more water molecules at this resolution, why only 20 were added? Space group: P32 2 1 (2 in 32 should be a subscript).

As most of the structures described are predictions by AlphaFold, the suggested interaction sites should be complemented by mutational or other experimental studies, which seems to be the case to some extent, though this could be written out more explicitly to make it clear to the reader for each case what was verified and how, if this is missing for any, it should be declared in limitations of the study. (Overall I think this might be good to mention that molecular structural details warrant future experimental studies to be fully verified). ipTM values are mentioned but i can't actually see them anywhere, would be nice to have those also.

Reviewer #4

(Remarks to the Author)

The manuscript of Anton-Plagaro et al. describe the outcome of an extensive proximity-labeling approach using tagged retromer and retriever subunits. The authors provide a data-rich resource of new components in proximity to both complexes and their interacting subunits, and identify in this context both Rab GEFs and GAPs as interacting proteins. Their data suggest that retromer controls Rab10 and Rab35, whereas retriever does not show similar interactions. Using a crystal structure of Vps29 with a peptide of the GEF DENND4C, as well as AlphaFold modelling, the authors generate selected mutations that disable the binding of the GEF and the GAP TBC1D3 and report defects in transport of selected cargoes. Their data reveal how retromer may generate sub domains on endosomes for cargo collection and sorting in cooperation with Rab GTPases.

This is an impressive and data-rich manuscript, which will be a great resource for the community. It is obvious that the authors have made tremendous advances via the BioID approach and demonstrate that they are able to carefully dissect interactions using state-of-the-art methodology. This is supported by their point-mutant analysis, which shows that they have identified interfaces with selected GEFs and GAPs and retromer. While I am impressed with their analysis, I am less enthused about their discussion and overall conclusions. The authors report on multiple interactions of retromer (and retriever) and disable binding with mutations, yet it becomes unclear how GEF and GAP interactions with retromer should then function together to ensure successful cargo selection and transport. I do not expect that the authors dissect this in this manuscript completely, but encourage them to improve their working model and their discussion. As it stands, readers will not understand, what we learned by these multiple interactions that are now described in their study.

Specific comments:

1. It is intriguing that the authors find interactions of retromer with both DENND4C and the GAP TBC1D13. It would be of interest to dissect this a bit further to understand the crosstalk between retomer and the Rab regulators. Do the authors believe that both GEF and GAP are bound to the same complex? Are these different complexes of retomer with GEF and GAP at different endosomal sites? How does the GEF localization fit with the Rab10 and Rab35 localization? Is the GAP TBC1D13 found on endosomal sites together with the GEF DENND4C?
2. In Figure 5C, the authors show GLUT1 Western blots. I am sorry, but I just see a blur in these gels, and it is hard to believe that this is a specific signal that the authors are following. Is there a better way to show the protein, potentially by deglycosylation prior to SDS-PAGE? I am also not sure how such a smear can be quantified. Please clarify.
3. Figure 7F shows many pieces of information with multiple GAPs and GEFs that may bind to retromer, but not retriever. If retromer binds to many different Rab regulators, how should this work? Different complexes at different sites? An additional model on the endosomal system would be more helpful than a molecular model.

Minor mistakes:

Line 501 – should be DENND4C?

Line 520 – missing verb “Similar dimerization WAS also observed in TBC1D5....”

Version 1:

Reviewer comments:

Reviewer #1

(Remarks to the Author)

The authors addressed my comments adequately and did a great job on the revision.

Reviewer #2

(Remarks to the Author)

The authors addressed my concerns well in the revision.

I have no further comments on the manuscript.

I think the revised version is suitable for publication.

Reviewer #3

(Remarks to the Author)

In my opinion the authors have sufficiently answered the questions to warrant acceptance of the manuscript.

Reviewer #4

(Remarks to the Author)

The authors addressed all my concerns. I consider this a very well conducted and valuable study to further understand retromer and retriever function.

Reviewer #1 (Remarks to the Author):

The manuscript by Antón-Plágaro et al. describes the endosomal proximity-dependent proteomics of Retromer and Retriever, their cargo adaptors, and an ESCRT-degradative sub-domain protein. By combining proximity-labeling proteomics, X-ray crystallography, and AlphaFold2, the authors provided a proteomic landscape of endosomal sorting sub-domains and identified the previously unappreciated interactions between Retromer and RAB10&RAB35 regulators.

This is a well-written, well-designed, and controlled study. All experiments were conducted at a high level and the presented results generally support the conclusions. The study describes a workflow for proteomics analysis of endosomal retrieval and recycling of integral cargo proteins and provides novel insight into the role of Retromer in neuroprotection. In my view, the content and quality of this study make it suitable for publication in Nature Communications. We thank the reviewer for their kind comments.

Line 118: I appreciate that the authors carefully generated stable cell lines (by knocking our endogenous genes and by selecting clones expressing as near to endogenous levels of the chimera proteins) and extensively validated the localization and function of each chimera. Also, the two negative controls (cytosolic BioID and TOM70-RFP-BioID) are a good addition. I also like that the authors mentioned/discussed known interacting proteins with baits that could not be captured in this study. Overall, preparation / optimization for BioID cell lines and following proteomic analysis were conducted at a high level. We thank the reviewer for these comments.

Figure 1-3: It would be hard for readers to follow the relationships between bait (BioID) proteins and their interactions. A cartoon(s) (like Figure 1B) or interaction networks must be very helpful to better understand an overview of the localization and topology of bait proteins and proximity-dependent interaction partners. We have thought carefully about the overview point. We fully appreciate that the data is complex and hence can be difficult to penetrate, and we have endeavoured to minimise the complexity wherever possible. We have performed a network analysis and visualised this across all the data through the dot-blots in Figure 3 (with the full data in Extended Figure 4). For example, those proteins that cluster as 'Transmembrane', those associated with 'Microtubules', etc. Our feeling is that this mode of data representation is the easiest for the reader. We have however added additional cartoons, derived from Figure 1B to each of Figures 1E-H and Figures 2A-C to focus the proteomic data back to the individual BioID-tagged protein. We hope these further aid understanding.

Line 159: Is "50 mM" biotin correct? Line 159: 50 μ M is the correct concentration, apologies.

Line 194: Mention how to correct batch effect for comparing different TMT measurements (if you performed). A full replicate of the experiment was run in each TMT batch, and the paired statistical approach was selected to account for the resulting batch effect.

Line 231: Could this be due to the relatively high-intensity signal of SNX27 in the negative control (cyto control), which made SNX27 less enriched in the experimental condition? A choice of negative control is sometimes a key to identifying bona fide interactions. This is an interesting point. We have now generated SNX27 tagged with TurboID (and separately UltraID) to provide time-resolved proximity proteomic labelling. This has revealed the SNX27-dependent biotinylation of Retromer, that we consider likely reflects ‘a highly dynamic, transient association between cargo-loaded SNX27 and the Retromer demarcated sub-domain during the handover of captured cargo’ (quote from lines 232-233 of manuscript). Our time-resolved analysis of SNX27, and more generally the organisation of the retrieval sub-domain, is a new and on-going study and hence we have not included these data in the revised manuscript.

Along the same line, BioID is a relatively slow reaction (~12 h) compared to APEX2/HRP (~min), TurboID (~10 min), AirID (~30 min). Please comment about pros&cons including the possibility of loss of transient interactions with BioID. Through lines 114-117 we discuss why we selected BioID1 for this initial proximity proteomic study. In “Limitations of the study” we highlight the relatively poor labelling kinetics of BioID1 and how more kinetically rapid labelling enzymes will allow time-resolved proximity proteins (see lines 706-713). We have edited the later to now include specific reference to APEX2/HRP, TurboID and AirID.

Line 891: Is FDR 5% at PSM, peptide, or protein level? Explain why you adopted the 5% FDR (at PSM level?), though most of the studies use 1% FDR. It is our view that for exploratory studies where substantial follow-up analysis and confirmation of hits is likely, a 5% FDR cutoff is more appropriate as it decreases the risk of highly biologically significant proteins being permanently lost from the dataset as false negatives. FDR scores are kept in the dataset throughout and are used to inform the selection of candidate proteins of interest on a protein-by-protein basis after statistical analysis.

Table: In the excel file, some functions remained (e.g., $-\text{LOG}_{10}([\text{Stats_X27/Cyto_T-Test}])$), keep only values. Apologies for this oversight, which has been corrected.

Deposit the proteomic raw files to a repository (e.g., PRIDE). The complete data set has been deposited at PRIDE with the dataset identifier PXD061918. This identifier has now been added to the revised text (see lines 961-963 in the Materials and Methods).

Reviewer #2 (Remarks to the Author):

The article presents a comprehensive map of the endosomal proximity proteome, offering insights into the molecular networks governing endosomal trafficking. By employing carefully designed BioID techniques, this study identifies the association of Retromer with various previously uncharacterized components. Notably, its interaction with Rab GTPase regulators, including DENND4 and TBC1Ds, was extensively validated through biochemical and cellular analyses, as well as crystallography and AlphaFold predictions.

This work is an excellent example of how AlphaFold models can be utilized to study protein-protein interactions. The proteomic data on the endosomal sorting system in this study could be a valuable resource and guide for future research on Retromer components. The manuscript is well-written and contains critical analyses and discussions.

I have no major criticisms of this manuscript but offer some minor suggestions: We thank this reviewer for their kind comments.

A significant portion of the protein-protein interaction analysis is based on AlphaFold predictions. Specifically, Figures 4 and 6 contain key interpretations of these predicted models. I suggest that the authors provide the AlphaFold models used for these figures as PDB or PyMOL files in the supplementary data, allowing readers to examine the models more closely. We have uploaded these AlphaFold models to ModelArchive, which is publicly available for readers to examine more closely. The ID numbers for the model submissions are provided below and are now included in the manuscript as Table S3 (see line 1046 for point of reference to this table).

Model	ModelArchive ID	DOI
Vps35_Vps29_DENND4A	ma-gpsxe	10.5452/ma-gpsxe
Vps35_Vps29_DENND4C	ma-kixo0	10.5452/ma-kixo0
Vps35_Vps29_DENND11	ma-xv6fg	10.5452/ma-xv6fg
Vps35_Vps29_TBC1D1	ma-mupfy	10.5452/ma-gpsxe
Vps35_Vps29_TBC1D4	ma-igov0	10.5452/ma-igov0
Vps35_Vps29_TBC1D5	ma-uf8h5	10.5452/ma-uf8h5
Vps35_Vps29_TBC1D13	ma-rkiek	10.5452/ma-rkiek
TBC1D1_homodimer	ma-ce92s	10.5452/ma-ce92s
TBC1D1_TBC1D4	ma-4yqbq	10.5452/ma-4yqbq
TBC1D1tail_TBC1D4tail	ma-xv4vj	10.5452/ma-xv4vj
TBC1D4_homodimer	ma-8wube	10.5452/ma-8wube
TBC1D5_homodimer	ma-qv4wj	10.5452/ma-qv4wj
Vps26A_Vps35_Vps29_DENND4C1to1400	ma-77fz1	10.5452/ma-77fz1
Vps26A_Vps35_Vps29_TBC1D5	ma-h364h	10.5452/ma-h364h
Vps29_Vps35_Vps26A_TBC1D13	ma-eazq0	10.5452/ma-eazq0
Vps35_Vps26A_Vps29_DENND4A1140to1223	ma-b9i08	10.5452/ma-b9i08
Vps35_Vps26A_Vps29_DENND4C1177to1252	ma-c4705	10.5452/ma-c4705
Vps35_Vps26A_Vps29_VARP	ma-0p47r	10.5452/ma-0p47r

The determination of the crystal structure and model building was well executed. However, in Table 3 on page 78, the Rmerge value for the highest resolution shell is very high, and the $I/\sigma(I)$ ratio is only 0.99. This suggests that the highest resolution shell contains no useful data, and the resolution limit has been overestimated. Additionally, the deposited electron density (ED) maps in the PDB (ID: 8VOD) do not show the typical quality expected for a 2.12 Å resolution dataset. To ensure accuracy, the data should be trimmed to a lower resolution where $I/\sigma(I)$ is greater than 2 or 3. We thank the reviewer for raising this question and agree to trim the resolution of the dataset. We have now truncated the data from 2.12 Å to 2.35 Å resolution and re-refined it. The

Rmerge value and the I/Sigma(I) ratio for the highest resolution shell is now 0.195 and 2.35 respectively.

For the convenience of the Editor and reviewer, we have also attached revised **Table 3** below:

Table 3. Summary of crystallographic structure determination statistics

Data collection statistics	mVps29 – hDENND4C₁₁₈₉₋₁₂₀₆
PDB ID	8VOD
Beamline	MX2
Space group	P 3₂ 2 1
Resolution (Å)	34.13 – 2.35 (2.43 – 2.35)
a, b, c (Å)	42.90, 42.90, 172.74
α, β, γ (°)	90.00, 90.00, 120.00
Data collection temp. (K)	100
Wavelength (Å)	0.95365
Total observations	16,359 (1,561)
Unique reflections	8,279 (785)
Completeness (%)	99.7 (100.0)
R _{merge} ⁺	0.028 (0.195)
R _{pim} *	0.028 (0.195)
CC1/2	0.998 (0.891)
<I/σ(I)>	9.87 (2.35)
Multiplicity	2.0 (2.0)
Molecule/asym	1
Refinement statistics	
R _{work} /R _{free} (%) ¶#	18.78/25.22
No. protein atoms	1590
Waters	26
Wilson B (Å ²)	49.33
Average B (Å ²) [^]	53.92
Protein	53.84
Water	48.12

rmsd bonds (Å)	0.008
rmsd angles (°)	1.07
Ramachandran plot:	
Favored/outliers (%)	95.38/0.51

Reviewer #3 (Remarks to the Author):

Comments on experimental structure description for Antón-Plágaro et al.:

while otherwise technically results appear to be correct, the methods paragraph for crystallography should be completely rewritten, the language needs to be checked (English) and some descriptions need a clarification: peptide complex crystallization what is meant by "3 molar excesses"? 3 molar = 3 M? Perhaps the writer should rather refer to the molar ratio of the two molecules. We express our sincere apologies for the inclusion of text from an earlier draft of the manuscript. We have now extensively edited this, and the sections below, to provide greater clarity.

For this specific section we have added the revised paragraph below (see lines 1007-1014):

"Crystallization and structure determination. Crystals of VPS29 – DENND4C1189-1206 peptide complex were grown by hanging drop vapor diffusion method at 20°C in a condition containing 0.1 M Tris-HCl pH 8.5, 0.2 M MgCl₂, 30% PEG4000. Best quality crystal was obtained by adding 3-fold molar excess of DENND4C1189-1206 peptide into the purified mVps29, reaching a final protein concentration of 12 mg/ml, with a protein-to-reservoir drop ratio of 2:3. Crystals grew after 10 days were cryoprotected in reservoir solution supplemented with 10% glycerol. X-ray diffraction data were collected at 100 K at the Australian Synchrotron MX2 beamline."

None of the text (grammar) below makes sense: "Best quality crystal was obtained by adding 3 molar excesses of DENND4C1189-1206 peptide into the purified mVps29 to a final concentration of 12 mg/ml with a protein-reservoir drop ratio of 2:3. Crystals grew after 10 days were cryoprotected in reservoir solution with additional 10% glycerol. X-ray diffraction data were carried out at the Australian Synchrotron MX2956 beamline at 100 K." This appears to be just incomplete sentences badly written. This paragraph should be checked by native / fluent speaker before it is possible to accept. (e.g data is not carried out anywhere, the data is collected). "crystal" should be "crystals" most likely. AutoXDS (ie XDS) is missing a reference. We thank the reviewer for the corrections. We have revised and checked this method section as mentioned above (see lines 1009-1014). We have checked the entire paragraph as mentioned above and added the appropriate reference next to the XDS (see line 1015).

Difference map and refinement description is wrong, or something strange was done. please correct the whole paragraph to exact and sensible description. You cannot use

F_o-F_c maps (alone) to build and inspect a model. We thank the reviewer for the comments. We agree, both 2F_o-F_c and F_o-F_c maps are needed for model refinement. Together with the modification described above, we have also edited this paragraph as shown below (see lines 1019-1021):

“The refinement was performed using Phenix (version 1.19.2), with inspection of the resulting model in Coot (version 0.8.9.2), guided by the 2F_o – F_c map and F_o – F_c difference map.”

Table 3 should include the wavelength (Å) used measurements, as well as the beam line used.

PDB ID is missing please add these, as a side note refinement should have included more water molecules at this resolution, why only 20 were added? Space group: P32 2 1 (2 in 32 should be a subscript). We have added the wavelength, the name of beamline used, PDB ID and the corrected space group notation in the revised Table 3.

Regarding the number of water molecules, we believe that, as suggested by reviewer 2, the data does not show a typical 2.12 Å resolution map. Instead, we believe the data only extends to 2.35 Å resolution (based on the I/Sigma(I) value). After careful re-refinement, we believe that 26 water molecules are typical for the Vps29 structure at this resolution.

For the convenience of the Editor and reviewer to follow, we have also attached revised Table 3 above.

As most of the structures described are predictions by Alphafold, the suggested interaction sites should be complemented by mutational or other experimental studies, which seems to be the case to some extent, though this could be written out more explicitly to make it clear to the reader for each case what was verified and how, if this is missing for any, it should be declared in limitations of the study. (Overall, I think this might be good to mention that molecular structural details warrant future experimental studies to be fully verified). ipTM values are mentioned but i can't actually see them anywhere, would be nice to have those also. We have included the ipTM values next to all the Alphafold models described in this manuscript including Figure 4A, 6A, S5A, S5B, S6A, S6C, S6E, S9A, S10B, S11, S12B and S12C.

Reviewer #4 (Remarks to the Author):

The manuscript of Anton-Plagaro et al. describe the outcome of an extensive proximity-labeling approach using tagged retromer and retriever subunits. The authors provide a data-rich resource of new components in proximity to both complexes and their interacting subunits, and identify in this context both Rab GEFs and GAPs as interacting proteins. Their data suggest that retromer controls Rab10 and Rab35, whereas retriever does not show similar interactions. Using a crystal structure of Vps29 with a peptide of the GEF DENND4C, as well as AlphaFold modelling, the authors generate selected mutations that disable the binding of the GEF and the GAP TBC1D3 and report defects in

transport of selected cargoes. Their data reveal how retromer may generate sub domains on endosomes for cargo collection and sorting in cooperation with Rab GTPases.

This is an impressive and data-rich manuscript, which will be a great resource for the community. It is obvious that the authors have made tremendous advances via the BioID approach and demonstrate that they are able to carefully dissect interactions using state-of-the-art methodology. This is supported by their point-mutant analysis, which shows that they have identified interfaces with selected GEFs and GAPs and retromer. We thank this reviewer for their kind comments.

While I am impressed with their analysis, I am less enthused about their discussion and overall conclusions. The authors report on multiple interactions of retromer (and retriever) and disable binding with mutations, yet it becomes unclear how GEF and GAP interactions with retromer should then function together to ensure successful cargo selection and transport. I do not expect that the authors dissect this in this manuscript completely, but encourage them to improve their working model and their discussion. As it stands, readers will not understand, what we learned by these multiple interactions that are now described in their study. This is extremely challenging, and we are hesitant to speculate too much at this stage. However, to provide some points for consideration we have included the following in the Discussion which we hope will suffice (lines 639 – 689):

“How are all these GEF and GAP interactions coordinated during Retromer’s function to ensure successful cargo selection and transport? This is a very challenging question that ultimately will require more targeted analysis utilising the newly acquired structural information coupled with knock-in technology to precisely disrupt protein:protein interactions, and high-speed, 4D multi-spectral live imaging of endogenously tagged Retromer, cargo, and individual RAB regulators to reveal the timeline of RAB regulation across cargo entry and exit through the retrieval sub-domain. That said, it is important to note that based on expression levels from OpenCell (<https://opencell.sf.czbiohub.org>), HEK293T cells express far more Retromer than Retriever (VPS35 at 800 nM compared with VPS35L at 72 nM), and that VPS29A is expressed at sufficiently high enough excess (approximately 1100 nM) to be present in both complexes without necessarily having to compete between the two. Moreover, the Retromer associated RAB regulators are expressed between 5-to-42-fold lower (DENND4A – 26 nM; VARP – 29 nM; TBC1D1 – 34 nM; DENND4C – 64 nM; TBC1D5 – 72 nM; TBC1D13 – 190 nM; TBC1D4 – 210 nM), establishing that the availability of VPS29 binding sites far exceeds the concentration of PL motif-containing proteins. The pseudo-helical array of Retromer arches therefore has the capacity to be heterogeneously labelled with RAB regulators and with the FAM21-containing WASH complex (FAM21A expressed at 180 nM).

The relative concentration of each individual regulator, their individual affinities derived from PL motif recognition and secondary recognition by VPS35, and additional low affinity binding to features of the local environment such as lipids, membrane geometry, F-actin and non-Retromer-interacting proteins, will provide the avidity that defines their enrichment and residency at the Retromer-demarcated retrieval sub-domain. Along

with their relative catalytic activity, this will define the local activation state of RABs and the association or disassociation of their functional effectors. The dynamic instability within the Retromer accessory protein network, is likely a feature in pathway progression and pathway fidelity that ensures directionality towards the successful biogenesis of a cargo-loaded transport carrier decorated with the appropriate RAB identity code, with post-translational modifications modifying and fine-tuning pathway progression by strengthening and weakening the avidity of Retromer network associations (Schmid and McMahon, 2007). The most simplified model to describe a highly dynamic and highly complex set of associations, is that the co-residence of TBC1D5 and TBC1D13 to a pseudo-helical Retromer coated tubule would ensure that membrane exiting the retrieval sub-domain has RAB7 in an inactive state (i.e. lacks late endosomal identity), and is devoid of active RAB35 to restrict association of its effector MICAL1 thereby preventing F-actin disassembly while simultaneously allowing ARF6 activation to aid recycling (Allaire et al., 2013; Frémont et al., 2017). Association of DENND4A/C and VARP would drive activation of RAB10 and RAB21 and the acquisition of compartment identity and effector recruitment necessary for cargo delivery into onward recycling compartments (Pei et al., 2023). As stated, this is certainly an oversimplification of a highly dynamic and complex set of regulatory steps and does not consider the potential distinct environment between SNX3-Retromer coated tubular profiles and RAB7-Retromer coated tubular profiles emerging from endosomes (e.g. binding of RAB7-GTP to VPS35 may restrict access to the secondary binding required by some regulators thereby ensuring they preferentially associate with SNX3-Retromer arrays). Neither does it consider the possibility of preferential in trans association of certain PL motif-containing accessory proteins such as VARP and FAM21 across pseudo-helical Retromer arches (Crawley-Snowdon et al., 2020; Guo et al., 2024; Romano-Moreno et al., 2024) over in cis association with membrane-associated monomeric Retromer. A considerable amount of additional work will be required to build a detailed mechanistic understanding of the interface between Retromer and these RAB regulators during endosomal cargo recycling.

Specific comments:

1. It is intriguing that the authors find interactions of retromer with both DENND4C and the GAP TBC1D13. It would be of interest to dissect this a bit further to understand the crosstalk between retromer and the Rab regulators. Do the authors believe that both GEF and GAP are bound to the same complex? As discussed above, we consider that higher-ordered Retromer assemblies potentially are heterogeneously decorated with these accessory proteins (see lines 670-671). The relative levels of GEFs versus GAPs are likely to be controlled through avidity-based interactions and by means of post-translational modifications. Together, these will establish a highly dynamic system to allow the controlled switching of RAB identity during the process of pathway progression.

Are these different complexes of retromer with GEF and GAP at different endosomal sites? Is the GAP TBC1D13 found on endosomal sites together with the GEF DENND4C? Our confocal analysis is consistent with TBC1D13 and DENND4C localising to Retromer decorated endosomes (see Figures 5A and 6H). At the resolution of our

imaging, we cannot conclude whether they associate with distinct endosomal sub-domains, this would require super-resolution and/or immuno-EM based analysis.

How does the GEF localization fit with the Rab10 and Rab35 localization? We have tried to visualise the relative localisation of Retromer with these RABs. Unfortunately, we have been unable to source suitable RAB10 and RAB35 antibodies for endogenous imaging of their localisation.

2. In Figure 5C, the authors show GLUT1 Western blots. I am sorry, but I just see a blur in these gels, and it is hard to believe that this is a specific signal that the authors are following. Is there a better way to show the protein, potentially by deglycosylation prior to SDS-PAGE? I am also not sure how such a smear can be quantified. Please clarify. We followed our previously published quantification methodology by applying a box around the entire GLUT1 signal for quantification of Odyssey acquired data (see e.g. Steinberg et al., Nat Cell Biol 2013). As quantification of GLUT1 is not straightforward, we also included quantification of other well documented Retromer cargo CTR1 and KIDINS220 to ensure that conclusions were derived from multiple cargo proteins (see Fig. 5E and 5F).

3. Figure 7F shows many pieces of information with multiple GAPs and GEFs that may bind to retromer, but not retriever. If retromer binds to many different Rab regulators, how should this work? Different complexes at different sites? An additional model on the endosomal system would be more helpful than a molecular model. We agree that these are extremely interesting and important questions. However, addressing them will require extensive additional work utilising endogenous tagging and multi-spectral high-speed imaging to view their dynamic relationships within the highly dynamic process of Retromer-mediated endosomal cargo sorting (we now state this in lines 640-645 of the Discussion). We are presently engineering appropriate cell systems (e.g. i3Neurons) and establishing collaborations to perform the required multi-spectral 4D imaging.

Minor mistakes:

Line 501 – should be DENND4C? This typo has been corrected.

Line 520 – missing verb “Similar dimerization WAS also observed in TBC1D5....” The grammar has been corrected.